# A Gamma-adapted subunit vaccine induces broadly neutralizing antibodies against SARS-CoV-2 variants and protects mice from infection

Lorena M. Coria [1,2] ✉, Juan Manuel Rodriguez[3,4], Agostina Demaria[1,2], Laura A. Bruno[1,2], Mayra Rios Medrano[1,2], Celeste Pueblas Castro[1,2], Eliana F. Castro[1,2], Sabrina A. Del Priore[3], Andres C. Hernando Insua[3,4], Ingrid G. Kaufmann[3], Lucas M. Saposnik[1,2], William B. Stone[5], Lineia Prado[1,2], Ulises S. Notaro[6], Ayelen N. Amweg[6], Pablo U. Diaz[6], Martin Avaro[7], Hugo Ortega[6], Ana Ceballos[8], Valeria Krum[3], Francisco M. Zurvarra[3,4], Johanna E. Sidabra[3], Ignacio Drehe[3], Jonathan A. Baqué[3], Mariana Li Causi[3], Analia V. De Nichilo[3,4], Cristian J. Payes[3], Teresa Southard[5], Julio C. Vega [9], Albert J. Auguste [5,10], Diego E. Álvarez[1,2], Juan M. Flo [3], Karina A. Pasquevich [1,2] & Juliana Cassataro [1,2] ✉

In the context of continuous emergence of SARS-CoV-2 variants of concern (VOCs), one strategy to prevent the severe outcomes of COVID-19 is developing safe and effective broad-spectrum vaccines. Here, we present preclinical studies of a RBD vaccine derived from the Gamma SARS-CoV-2 variant adjuvanted with Alum. The Gamma-adapted RBD vaccine is more immunogenic than the Ancestral RBD vaccine in terms of inducing broader neutralizing antibodies. The Gamma RBD presents more immunogenic B-cell restricted epitopes and induces a higher proportion of specific-B cells and plasmablasts than the Ancestral RBD version. The Gamma-adapted vaccine induces antigen specific T cell immune responses and confers protection against Ancestral and Omicron BA.5 SARS-CoV-2 challenge in mice. Moreover, the Gamma RBD vaccine induces higher and broader neutralizing antibody activity than homologous booster vaccination in mice previously primed with different SARS-CoV-2 vaccine platforms. Our study indicates that the adjuvanted Gamma RBD vaccine is highly immunogenic and a broad-spectrum vaccine candidate.

Severe Acute Respiratory Syndrome Coronavirus 2 (SARS-CoV-2) has infected over 777 million people worldwide and resulted in more than 7 million deaths (WHO. World Health Organization, Weekly Operational Update on COVID-19). The vaccines authorized for emergency use or fully approved are safe and highly effective against severe disease[1–5]. However, it has been reported that vaccine-induced protection against symptomatic SARS-CoV-2 infection wanes over time[6,7]. Additionally, as COVID-19 pandemic progresses several variants of concern (VOCs) have emerged, including B.1.351 (Beta), B.1.617.2 (Delta), P.1 (Gamma), B.1.1.529 (Omicron BA.1) and its subvariants (BA.2.12.1, BA.4/BA.5, BQ.1 and XBB and its descendants). Omicron lineages have been reported to be more resistant to

neutralization by vaccine-elicited antibodies and, in some cases are more transmissible than previous VOCs[8–12].

Waning of vaccine-induced immunity and antibody evasive virus variants create the need for new vaccine strategies that can induce more potent, durable, and broader immune responses to enhance protection against SARS-CoV-2. Additional booster doses of current vaccines have been implemented around the world. First approved vaccines were mainly directed against the spike protein of the prototype Wuhan-1 SARS-CoV-2 strain therefore their efficacy against certain VOCs is limited[11]. Therefore, vaccines are being adapted to variants as a strategy to improve vaccine effectiveness. While current vaccines may be effective at reducing severe disease to existing VOCs, variant-specific antigens, whether in a mono- or multivalent-vaccine, may be required to induce optimal immune responses and reduce infection against emerging variants.

Most of the variant-adapted vaccines boosters were bivalent or monovalent containing beta and Ancestral spike antigens besides Omicron[13–16]. Interestingly, the Beta variant originated in South Africa, while the Gamma variant originated in Brazil a few months later. Until now, there are few studies using a Gamma variant adapted vaccine[17–19] but recently, we have reported a phase 1 study of a Gamma-adapted receptor binding domain (RBD) subunit vaccine that induced a broad immune response against different VOCs[20].

In this work, preclinical studies showed that Gamma-adapted RBD vaccine is more immunogenic than the Ancestral RBD vaccine in terms of inducing broader neutralizing antibodies and higher antigen (Ag) specific cellular immune responses. Thus, the mechanisms underlying the superior immunogenicity of the Gamma RBD vaccine were explored. The Gamma adapted vaccine immunogenicity was evaluated after a primary two-dose vaccine schedule or after a heterologous booster of different anti-SARS-CoV-2 vaccine platforms including the non-replicating adenovirus vaccine ChAdOx1-S (Oxford AstraZeneca), the mRNA vaccine BNT162b2 (Pfizer) and the inactivated SARS-CoV-2 vaccine BBIBP-CorV (Sinopharm).

## Results

### Design and production of RBD antigens

To compare immune responses elicited with Ancestral or Gamma derived RBD, two formulations were developed using RBD homodimers from the Ancestral (Wuhan-Hu-1) or Gamma variant. Antigens comprise single-chain dimeric repeats of the RBD protein from amino acids 319 R to 537 K. The Gamma RBD antigen includes the mutations described in the Spike protein of the Gamma SARS-CoV-2 VOC: K417T; E484K and N501Y (P.1/501Y.V3). High-productivity clones were generated under the genetic background of a CHO-S (Chinese Hamster Ovary) cell line in high-density suspension cell cultures. Then, proteins were purified, and antigen purity confirmed by SDS-PAGE and Western Blot (Fig. 1a, b). Endotoxins, host cell proteins and residual host DNA were examined and complimented the quality requirements for GMP biological products. Comparison of the binding affinity of Gamma and Ancestral RBD to hACE2 receptor was performed by ligand-receptor binding ELISA (Fig. 1c). Both antigens bind to hACE2 at comparable levels with a slight greater binding affinity of Gamma RBD. Purified antigens were then absorbed to aluminum hydroxide (Alum) to generate the final formulations.

### Gamma RBD outperforms ancestral RBD antigen vaccine formulation on elicited antibody responses

Immunogenicity of vaccine formulations was assessed in BALB/c mice after intramuscular (i.m.) immunization in a two-dose vaccine schedule using 10 μg of each antigen adsorbed to aluminum hydroxide (Fig. 2a). RBD (Gamma, Ancestral or Omicron BA.4/5)-specific IgG was evaluated by ELISA in sera of immunized animals. Both vaccine formulations induced high levels of IgG antibodies against the Ancestral and Gamma RBD antigens (Fig. 2b). Of note, at day 14 post first dose, the Gamma antigen formulation induced significantly higher anti- Gamma or Ancestral RBD IgG responses than the Ancestral RBD-based vaccine (Fig. 2c). At day 42, the Gamma antigen formulation induced significantly higher anti-Gamma or Omicron BA.4/5 RBD IgG responses than the Ancestral RBD-based vaccine (Fig. 2c).

Neutralizing antibody activity was evaluated by a live virus assay using Ancestral or Gamma SARS-CoV-2 variants. Immunization with the formulation containing Gamma RBD plus Alum elicited high neutralizing antibody titers against both variants at all time points (Fig. 2d). In contrast, Ancestral RBD antigen induced higher neutralization titers against Ancestral SARS-CoV-2 than against Gamma variant until day 35 post prime immunization. Besides, vaccine formulation containing Gamma RBD also induced a higher antibody response with broader neutralizing activity than immunization with a formulation containing recombinant Ancestral Spike 2 P as antigen plus Alum (Supplementary Fig. 1).

To evaluate the mechanisms underlying the superior immunogenicity of the Gamma RBD based vaccine, specific-B cells and plasmablasts were evaluated at day 28 post second dose in splenocytes

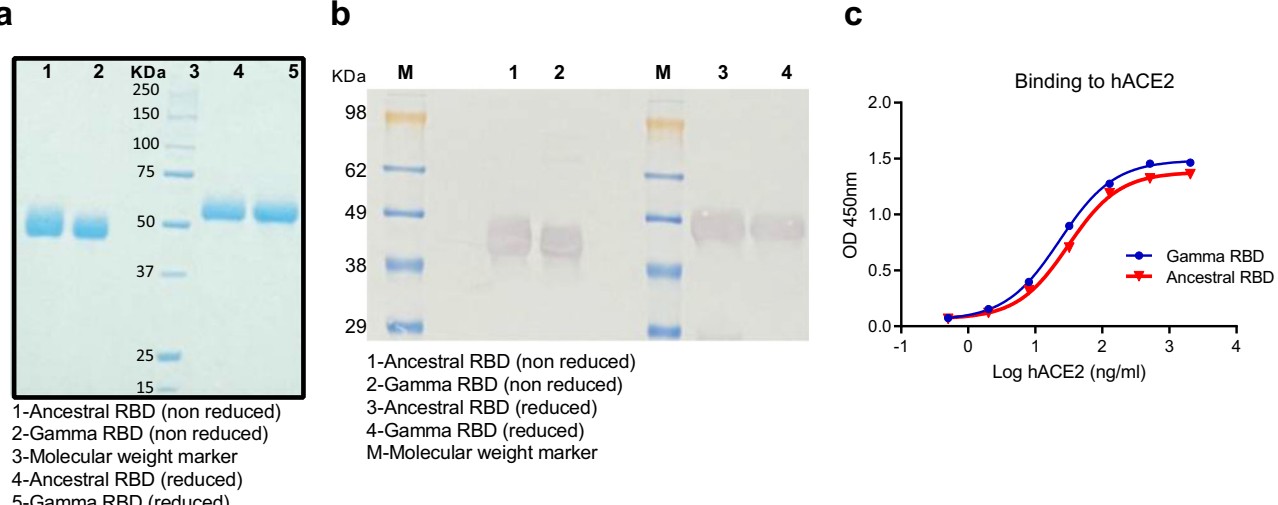

**a**

1-Ancestral RBD (non reduced)
2-Gamma RBD (non reduced)
3-Molecular weight marker
4-Ancestral RBD (reduced)
5-Gamma RBD (reduced)

**b**

1-Ancestral RBD (non reduced)
2-Gamma RBD (non reduced)
3-Ancestral RBD (reduced)
4-Gamma RBD (reduced)
M-Molecular weight marker

**c**

Binding to hACE2

**Fig. 1 | Characterization of RBD antigens. a** Non-reduced (nr) and reduced (r) SDS-PAGE migration profiles of Ancestral and Gamma RBD are shown. **b** Western blot. Non-reduced and reduced RBD antigens were detected using a rabbit polyclonal anti-RBD antibody. **c** Binding affinity of Gamma (blue circles) and Ancestral (red triangles) RBD proteins to immobilized human ACE2 receptor by ligand-receptor binding ELISA. Source data are provided as a Source Data file.

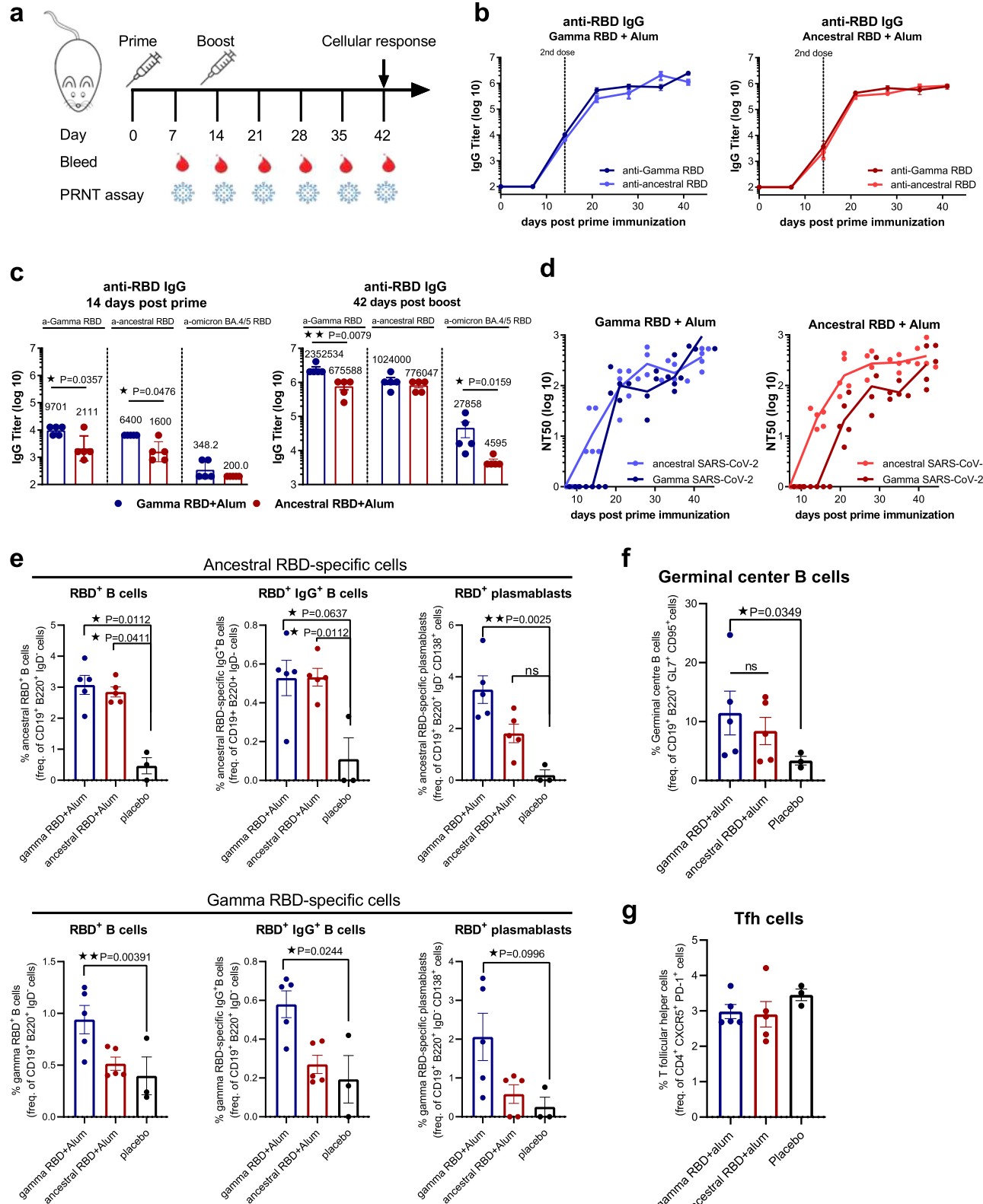

from immunized animals. Gamma RBD + Alum formulation increased the percentage of Ancestral or Gamma RBD-specific total B cells (B220⁺ CD19⁺ IgD⁻ RBD⁺), IgG⁺ B cells (B220⁺ CD19⁺ IgD⁻ IgG⁺ RBD⁺) and plasmblasts (B220⁺ CD19⁺ IgD⁻ CD138⁺ RBD⁺) while the vaccine containing the Ancestral version of RBD induced only Ancestral RBD-specific B cells and plasmablasts (Fig. 2e and Supplementary Fig. 2).

Germinal center (GC) B cells were also evaluated in spleens of immunized mice. Gamma RBD based-vaccine increased the frequency of GC cells (B220⁺ CD19⁺ IgD⁻ CD95⁺ GL7⁺ cells) compared to placebo group while no significant differences were found compared to the Ancestral RBD formulation (Fig. 2f). No differences in the frequency of T follicular helper cells (Tfh) were found between immunized and placebo groups (Fig. 2g and Supplementary Fig. 2). These results

**Fig. 2 | Gamma RBD-adapted vaccine induces a high and broad antibody response and specific-B cells. a** Immunization protocol scheme. BALB/c mice were immunized on day 0 and day 14 via i.m. with: Ancestral RBD + Alum (red) or Gamma RBD + Alum (blue). Serum samples were obtained at indicated time points for ELISA and neutralization assays. **b** Kinetics of RBD-specific IgG endpoint titers in sera of immunized animals by ELISA. Points are means ± SEM. $n = 5$ mice per group. **c** RBD-specific (Gamma, Ancestral or Omicron BA.4/5) IgG titers in sera of immunized animals at days 14 and 42 post prime immunization. Error bars are GMT ± SD. GMT values are shown above the bar. $n = 5$ mice per group. Two-sided Mann Whitney test. *$p < 0.05$, **$p < 0.01$. Exact $P$ values are shown. **d** Gamma SARS-CoV-2 (circles) or Ancestral SARS-CoV-2 (triangles) Neutralizing-antibody titers in sera were determined by authentic PRNT SARS-CoV-2 assay for each group of vaccinated mice at different time points. The lines represent the geometric mean (GMT) of all data points. $n = 5$ mice per group. Titers correspond to the 50% of virus neutralization (NT50). **e** BALB/c mice were immunized at day 0 and day 14 via i.m. with: Ancestral RBD + Alum ($n = 5$ mice, blue circles), Gamma RBD + Alum ($n = 5$ mice, red circles) or placebo ($n = 3$ mice, black circles). Flow cytometry analysis of different B-cell populations in spleen of vaccinated mice were performed using anti-CD19, anti-B220, anti-IgD, anti-IgG, anti-CD138, anti-GL7, and anti-CD95 antibodies. Specific-B cells were determined by binding to fluorescent RBD (Ancestral and Gamma versions). Results are presented as percentage of RBD+ B cells (CD19+ B220+ IgD− RBD+), RBD+ IgG+ B cells (CD19+ B220+ IgD− IgG+ RBD+) or RBD+ plasmablasts (B220+ CD19+ IgD− CD138+ RBD+). Bars are means ± SEM. Kruskal Wallis test. *$p < 0.05$, **$p < 0.01$. Exact $P$ values are shown. **f** BALB/c mice were immunized at day 0 and day 14 via i.m. with: Ancestral RBD + Alum ($n = 5$ mice, red circles), Gamma RBD + Alum ($n = 5$ mice, blue circles) or placebo ($n = 3$ mice, black circles). Frequency of germinal center B cells (B220+ CD19+ GL7+ CD95+) in each group of vaccinated mice. Bars are means ± SEM. Kruskal Wallis test. *$p < 0.05$, ns (not significant). Exact $P$ value is shown. **g** BALB/c mice were immunized at day 0 and day 14 via i.m. with: Ancestral RBD + Alum ($n = 5$ mice, red circles), Gamma RBD + Alum ($n = 5$ mice, blue circles) or placebo ($n = 3$ mice, black circles). Frequency of T follicular helper cells (CD4+ CXCR5+ PD-1+) in spleen of vaccinated mice. Bars are means ± SEM. Kruskal Wallis test. Results are representative of three independent experiments. Source data are provided as a Source Data file.

suggest that the superior antibody response of the Gamma adapted vaccine in comparison with Ancestral can be supported by a higher frequency of specific-B cells that could be involved in the generation of high avidity antibodies.

## Gamma RBD adjuvanted formulation induces long-lasting antibody responses with broad neutralizing activity against different VOCs

Neutralizing antibody (nAb) activity against different VOCs was evaluated one month after second immunization (42 days post prime vaccination). The Gamma RBD-based vaccine elicited antibodies with high neutralization capacity against Ancestral (GMT 397.7) and Gamma variant (GMT 418.1). Interestingly, these antibodies exhibited cross-neutralization capacity against Alpha (GMT 60.5), Delta (GMT 124.5), Omicron BA.1 (GMT 101.9) and Omicron BA.5 variants (GMT 62.4) (Fig. 3a). In comparison, the Ancestral RBD formulation induced similar levels of nAbs against Ancestral (GMT 430.1), Gamma (GMT 218.4), Alpha (GMT 46.7) VOC but significantly lower levels against Delta (GMT 23.7), Omicron BA.1 (GMT 23.7) and Omicron BA.5 (GMT 5.9) VOC (Fig. 3a).

Antigenic cartography was used to explore the different neutralization profiles elicited by the RBD antigens employed. Antigenic maps were made separately for each vaccine using the nAbs titers at day 28 after the second immunization. For the Gamma RBD-adapted formulation, data were more tightly clustered around the Ancestral and Gamma variant (Fig. 3b left panel) in comparison to the Ancestral RBD vaccine (Fig. 3b, right panel). Antigenic distances between Delta, Omicron BA.1 and Omicron BA.5 VOC and the vaccines homologous variants (Ancestral or Gamma) were lower for the vaccine containing the Gamma RBD antigen than the Ancestral RBD formulation (Supplementary Table 1).

Binding antibody responses were evaluated over longer periods after mice immunization revealing that anti-RBD IgG titers remained high up to 253 days (more than 8 months) post prime immunization (Fig. 3c). Additionally, neutralization capacity of antibodies elicited after immunization was evaluated using different VOCs in a live virus PRNT assay at this time point. Interestingly, at day 253, there was not a significant variation in the nAbs titers against the VOCs compared to day 42 (Fig. 3d). Plasma cells (PCs) derived from germinal centers secrete high-affinity antibodies required for long-term serological immunity. Analysis of long-lived plasma cells (LLPCs) in the bone marrow of immunized mice at day 42 dpp indicates that the frequencies of total (B220− CD138+) and isotype-switched RBD-specific LLPCs (B220− CD138+ IgG+ RBD+) were higher in Gamma subunit vaccine vaccinated mice compared to placebo group (Supplementary Fig. 3). These results further explain the induction of a broader and long-lasting nAb response by the Gamma RBD adjuvanted formulation.

To explore the mechanisms that could be involved in the superior antibody response of the Gamma antigen, the primary sequence of RBD of Spike protein from Ancestral or Gamma SARS-CoV-2 were scanned by the IEDB server to predict linear B cell epitopes. Among all the predicted epitopes, those exposed on the surface of S protein with a length between 5 and 25 residues were selected (Supplementary Table 2). Vaxijen 2.0 was used to calculate antigenicity scores. Among the 5 selected epitopes, the peptide sequence starting at position 496 showed the highest score in both variants (antigenic scores were 0.7632 and 0.8457 for Ancestral and Gamma antigen respectively, Supplementary Table 2). Interestingly, this epitope in the Gamma RBD sequence contains threonine for asparagine substitution in residue 501 (N501Y). In addition, Disco tope analysis of discontinuous epitopes revealed that residues 500 to 506 can be considered as a potential discontinuous epitope in both antigens. Thus, this residue substitution could contribute to the superior immunogenicity of the Gamma RBD formulation.

## Gamma RBD-adapted vaccine elicits Ag-specific mixed Th1-Th2 immune responses

Cellular immune responses were analyzed in splenocytes from immunized animals by intracellular flow cytometry. The Gamma RBD-based vaccine induced a significant increase in the frequencies of Gamma RBD-specific IFN-y-, TNF-α- and IL-2-producing CD4+ T cells after restimulation with the Gamma antigen (while the ancestral version of the vaccine failed to induce IFN-y in CD4+ T cells but could induce TNF-α and IL-2 after restimulation Fig. 4a, upper panel). Both vaccines have a similar performance at inducing antigen specific CD8+ T cells (Fig. 4a, lower panel). Representative dot plot graphs are shown in Supplementary Fig. 4. Also, RBD-specific CD4+ and CD8+ T cell immune responses were detected in splenocytes from C57BL/6 mice immunized with the Gamma version of the vaccine (Supplementary Fig. 5).

In addition, analysis of polyfunctional T cells showed that responses elicited by the Gamma RBD based vaccine had a predominance of single positive TNF-α+ cells and specific double-positive (IFN-y+ and TNF-α+) cytokine producing CD4+ T cells (Supplementary Fig. 6). These results demonstrated that vaccine containing Gamma RBD antigen induces a superior Ag-specific Th1 cellular immune response than RBD derived from the Ancestral SARS-CoV-2.

Evaluation of Th cytokine profile was performed by multiplex assay in supernatants from splenocytes of immunized mice stimulated with Ancestral or Gamma RBD antigen. Results revealed that the Gamma RBD vaccine induced cytokines associated with Ag-specific Th1 andTh2 responses with production of IFN-y, TNF-α, IL-2, IL-6, IL-10, IL-4 and IL-5 upon Ag stimulation in comparison to unstimulated cells (Fig. 4b). After immunization with the Ancestral vaccine,

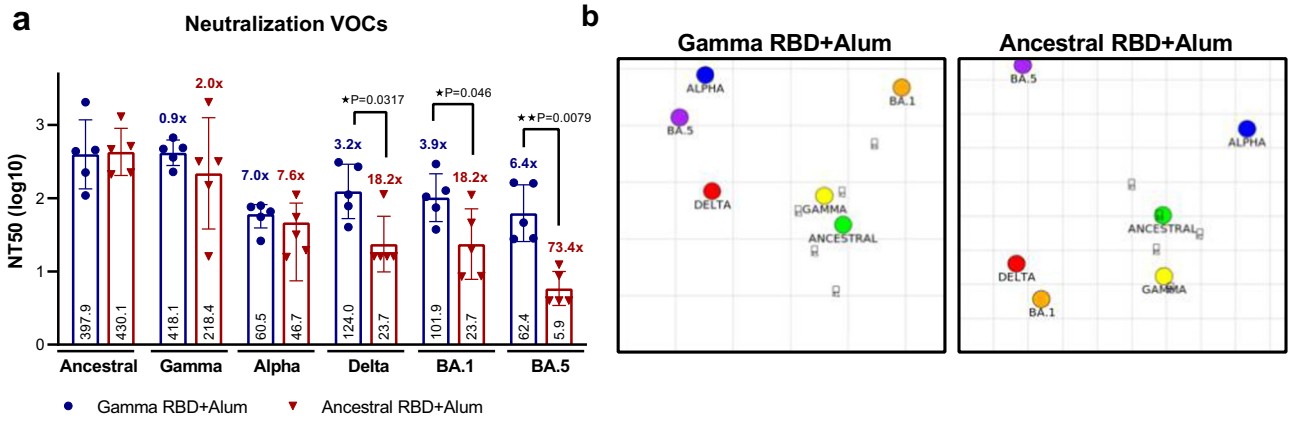

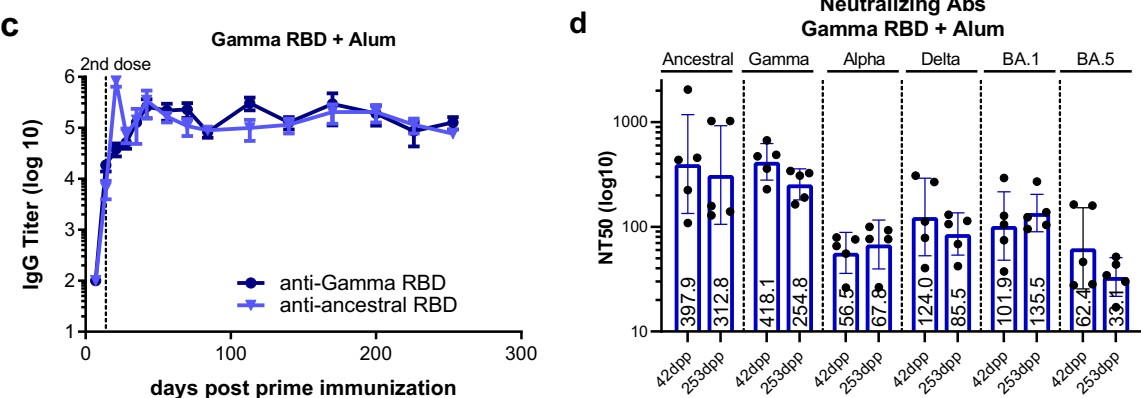

**Fig. 3 | Gamma RBD plus Alum vaccinated mice induce a long-lasting antibody response with a broader neutralization activity.** BALB/c mice were immunized at day 0 and day 14 via i.m. with: Gamma RBD + Alum (blue circles) or Ancestral RBD + Alum (red triangles). **a** Neutralizing antibodies titers in serum of vaccinated mice against Ancestral SARS-CoV-2 and different VOCs (Alpha (B.1.1.7), Gamma (P.1) Delta (B.1.617.2) and Omicron (BA.1 and BA.5). Neutralization titer was defined as the serum dilution that reduces 50% the cytopathic effect (NT50). Bars are GMT ± SD. GMT values are shown inside the bar. $n = 5$ mice per group. Fold-reduction against Ancestral virus titers is shown above the bar followed by an x. Two-sided Mann Whitney test. *$p < 0.05$, **$p < 0.01$. Exact p values are shown. Representative of three experiments. **b** Antigenic maps were generated from the neutralizing antibody titers in serum after the second dose of the Gamma RBD + Alum (left panel) or Ancestral RBD + Alum (right panel). Green, yellow, blue, red,

orange and purple circles correspond to the Ancestral, Gamma, Alpha, Delta, Omicron BA.1, and Omicron BA.5 variants, respectively. Each grid square corresponds to a two-fold dilution in the neutralization assay. The antigenic distance is interpretable in any direction. **c** Kinetics of RBD-specific (Ancestral or Gamma) IgG endpoint titers in sera of immunized animals by ELISA until 253 days post prime immunization (dpp). Points are means ± SEM. $n = 5$ mice per group (**d**). Neutralizing-antibody titers against Ancestral SARS-CoV-2 and different VOCs (Alpha (B.1.1.7), Gamma (P.1) Delta (B.1.617.2) and Omicron (BA.1 and BA.5) were evaluated at 42 and 253 dpp in both immunized groups. Neutralization titer was defined as the serum dilution that reduces 50% the cytopathic effect (NT50). Bars are GMT ± SD. GMT values are shown inside the bar. $n = 5$ mice per group. Results are representative of three independent experiments. Source data are provided as a Source Data file.

predominantly Th2 related cytokines were significantly produced (IL-2, IL-6, IL-13, IL-4 and IL-5) in comparison to unstimulated cells (Fig. 4b). Of note, significant differences between vaccinated groups were only found in IL-5 and IL-13 production. Interestingly, Th17, Th9 and Th22 related cytokines (IL-17A/F, IL-9 and IL-22) were induced by the Gamma RBD + Alum formulation.

These results demonstrated that both vaccines formulations elicited T cell immune responses with induction of Ag specific cytokine producing CD4$^+$ and CD8$^+$ T cells.

Analysis of immunodominant T cell restricted epitopes present in the Gamma and Ancestral RBD sequences was performed using netMHCpan 4.1 and netMHCpan 3.2. MHCI and MHCII restricted epitopes of 9 residues in length from the two more common mouse haplotypes (d and b) were selected and scored by Vaxijen 2.0. Results indicated that substitution N501Y present in the Gamma RBD sequences generated 3 antigenic epitopes restricted to the I-A$^b$ MHCII haplotype (Supplementary Table 3). These epitopes are not present in the Ancestral RBD sequence. No new epitopes or changes in the score values were found in the I-A$^d$ or I-E$^d$ MHCII haplotypes (BALB/c strain).

Identification of CTL epitopes revealed that new antigenic epitopes carrying the E484K and N501Y were generated in the Gamma sequence in both mouse haplotypes (H-2K$^{b/d}$ and H-2D$^{b/d}$, Supplementary Table 4). Experimentally, vaccinated mice with the Gamma RBD adjuvanted vaccine elicited RBD-specific CD4$^+$ and CD8$^+$ T cell responses in both mouse strains.

### Gamma RBD-based vaccine confers protection against intranasal SARS-CoV-2 challenge

To determine the vaccine's ability to protect against SARS-CoV-2 infection, a K-18-hACE2 transgenic mouse model was used[21,22]. Transgenic mice of both sexes were immunized on days 0 and 14 with Ancestral RBD plus alum, Gamma RBD plus alum, BNT162b2 bivalent vaccine or placebo as control. Two weeks later, mice were intranasally inoculated with $2 \times 10^4$ PFU of Omicron BA.5 SARS-CoV-2. High levels of viral RNA were detected in lung homogenates from placebo group at 3 days post infection (dpi) while in all vaccinated animals the viral genome copies were significantly reduced (Fig. 5a). However, at 5 dpi high viral gene copies were observed at the placebo but only the

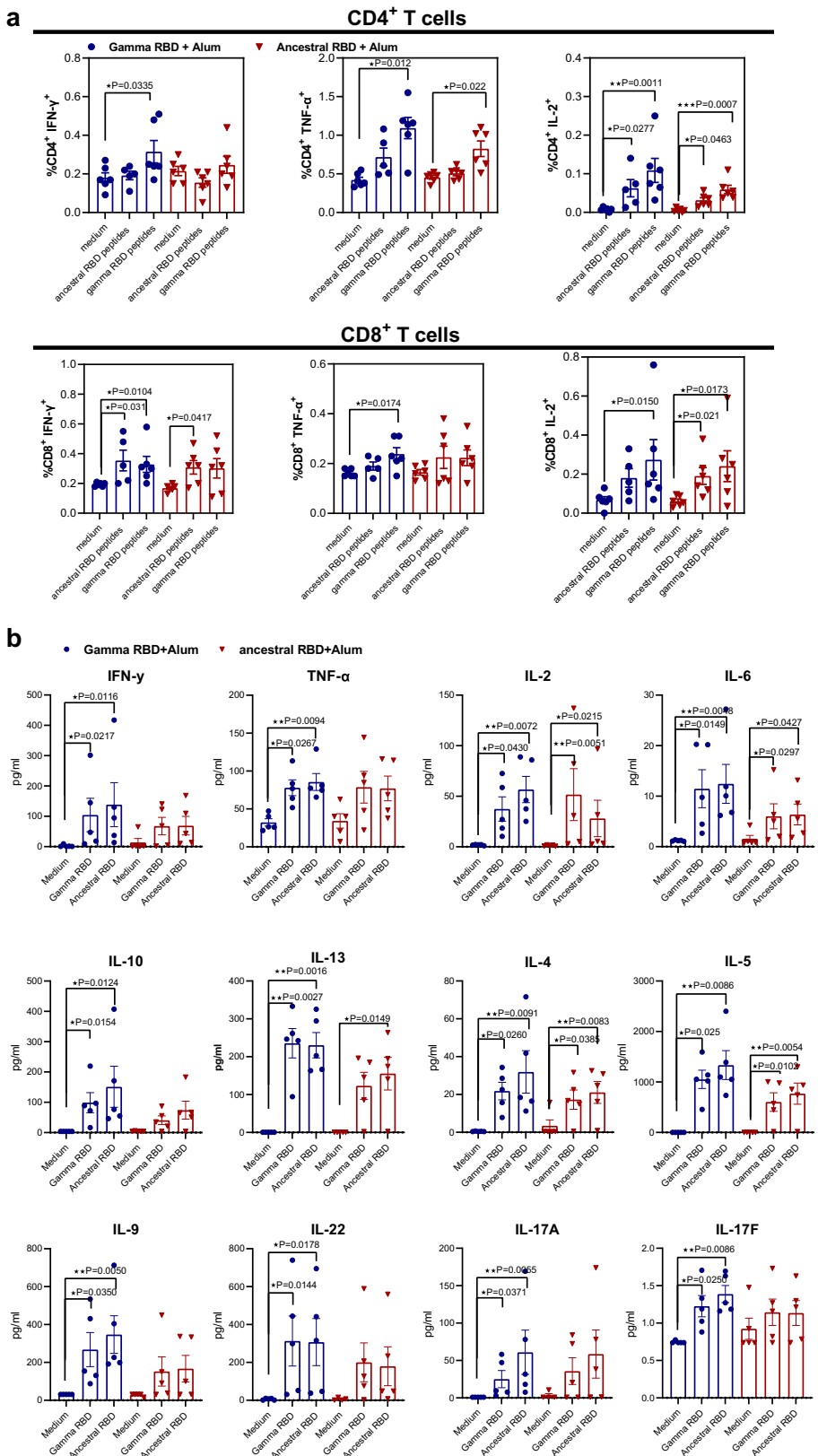

Gamma RBD plus Alum and bivalent mRNA vaccinated groups had reduced viral RNA at lung tissue (Fig. 5b). In agreement to the reduced pathogenicity reported for this virus variant, infected animals did not lose weight neither succumb to infection (Supplementary Fig. 7).

Histopathological analyses of lungs from infected mice of placebo group showed moderate focal to multifocal interstitial pneumonia,

peribronchial and perivascular infiltration with areas of moderate type II pneumocyte hyperplasia, hemorrhage, and diffuse edema (Fig. 5b, c). Bronchioles were surrounded by inflammatory cells, mainly neutrophils and mononuclear cells. Also, thickened alveolar walls and reduced alveolar spaces, with loss of epithelial cells and patchy alveolar edema, was observed at 5 dpi. The cumulative pathology score

**Fig. 4 | Gamma and ancestral RBD formulated with Alum induce T cell responses in mice.** BALB/c mice were immunized at day 0 and day 14 via i.m. with: Gamma RBD + Alum (blue circles) or Ancestral RBD + Alum (red triangles) and 28 days after T cell responses were evaluated. **a** Splenocytes were stimulated with complete medium or a peptide pool derived from RBD (Gamma and Ancestral) and then brefeldin A was added. Afterward, cells were harvested and stained with specific Abs anti-CD8, and anti-CD4, fixed, permeabilized, and stained intracellularly with anti−IFN-γ, TNF-α and anti-IL-2. Results are presented as percentage of cytokine-producing T cells. Bars are means ± SEM. $n = 6$ mice per group. Kruskal Wallis test. *$p < 0.05$, **$p < 0.01$, ***$p < 0.001$. Exact $P$ values are shown. Representative of two experiments. **b** Evaluation of secreted cytokines in the supernatants from stimulated splenocytes by flow cytometry. Bars are means ± SEM. $n = 5$ mice per group. Kruskal Wallis test. *$p < 0.05$, **$p < 0.01$. Exact $P$ values are shown. Representative of two experiments. Source data are provided as a Source Data file.

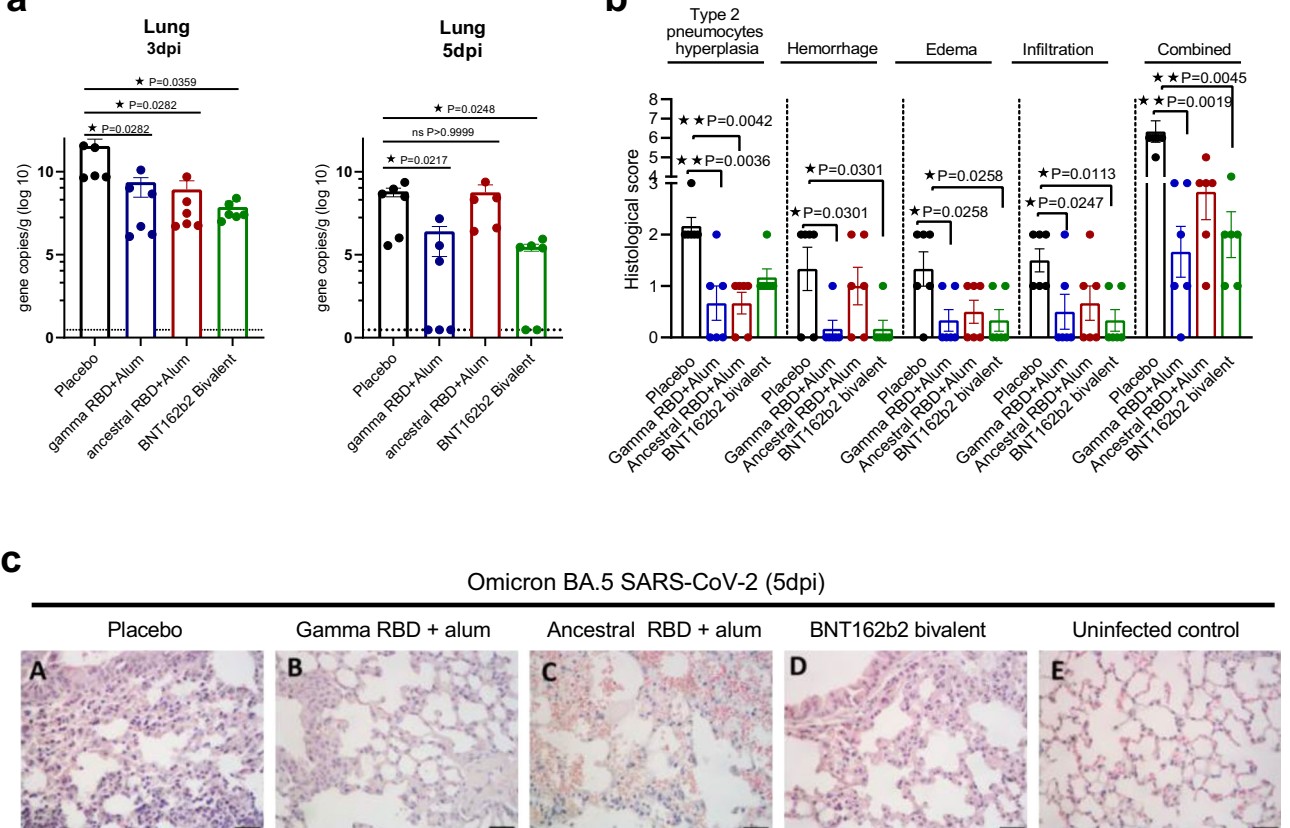

**Fig. 5 | Vaccination with Gamma RBD + Alum protects K18-hACE2 transgenic mice against intranasal Omicron BA.5 SARS-CoV-2 infection.** Mice were i.m. inoculated on days 0 and 14 with: (i) placebo, (ii) Gamma RBD + Alum, (iii) Ancestral RBD + Alum or (iv) bivalent BNT162b2. $n = 12$ mice per group. Two weeks following immunization, mice were intranasally infected with $1 \times 10^4$ PFU of Omicron BA.5. **a** Viral RNA copies were determined by RT-qPCR in lung homogenates at 3 and 5 dpi ($n = 6$ mice per time point). Bars are means ± SEM. Kruskal Wallis test. *$p < 0.05$.

Exact $P$ values are shown. **b** Pathological scores according to the severity of histological parameters were calculated for each group of mice at 5 dpi ($n = 6$ mice per group). Scores of all parameters were summed (Combined). Bars are means ± SEM. Kruskal Wallis test. *$p < 0.05$, **$p < 0.01$. Exact $P$ values are shown. **c** Hematoxylin and eosin (H&E) staining of the lung sections at 5 dpi. Placebo (A), vaccinated mice (B–D) and an uninfected animal (E, naive) are shown. Scale bars, 50 μm. Representative of two experiments. Source data are provided as a Source Data file.

which included the sum of all the pathological findings scores demonstrated moderate to severe histological changes in the placebo group. Histological analysis of lungs obtained from vaccinated mice showed a significant reduction in lung pathology score at 5 dpi (Fig. 5b). Vaccinated mice exhibited only a limited interstitial pneumonia with weak inflammatory infiltrate and hemorrhage, with significantly lower severity in animals immunized with Gamma RBD vaccine or the bivalent BNT162b2 vaccine (Fig. 5c).

In another independent experiment, placebo and Gamma vaccinated K-18-hACE2 mice were challenged intranasally with $2 \times 10^5$ PFU of Wuhan SARS-CoV-2 and monitored daily for body weight changes and survival. Immunized mice did not show significant weight loss during the experiment and 66% of the immunized mice survived at 9 dpi whereas all mice in placebo group succumbed to infection (Supplementary Fig. 8a, b). High infectious SARS-CoV-2 titers were observed in the lungs and brains of the placebo immunized mice at 5 dpi while low virus titers were detected in lungs and brains of animals vaccinated with Gamma RBD + Alum (Supplementary Fig. 8c). Histopathological changes were analyzed in lung tissue sections and scored according to the severity of the changes. In the placebo group, more significative changes were found in the fibrillar eosinophilic material in the alveolar spaces (edema) and cellular infiltrates in the alveolar septa. Mild or no changes were found in the plump epithelial cells lining the alveolar septa (type 2 pneumocyte hyperplasia) and infiltration of macrophages in the alveolar spaces. Vaccinated mice presented signs of alveolar cellularity but reduced edema in the lungs. No infiltration or pneumocytes hyperplasia was found in this group (Supplementary Fig. 8d, e).

Thus, immunization with the Gamma RBD-based vaccine induced protection against Ancestral and Omicron BA.5 variant in a mouse model of severe COVID-19 disease.

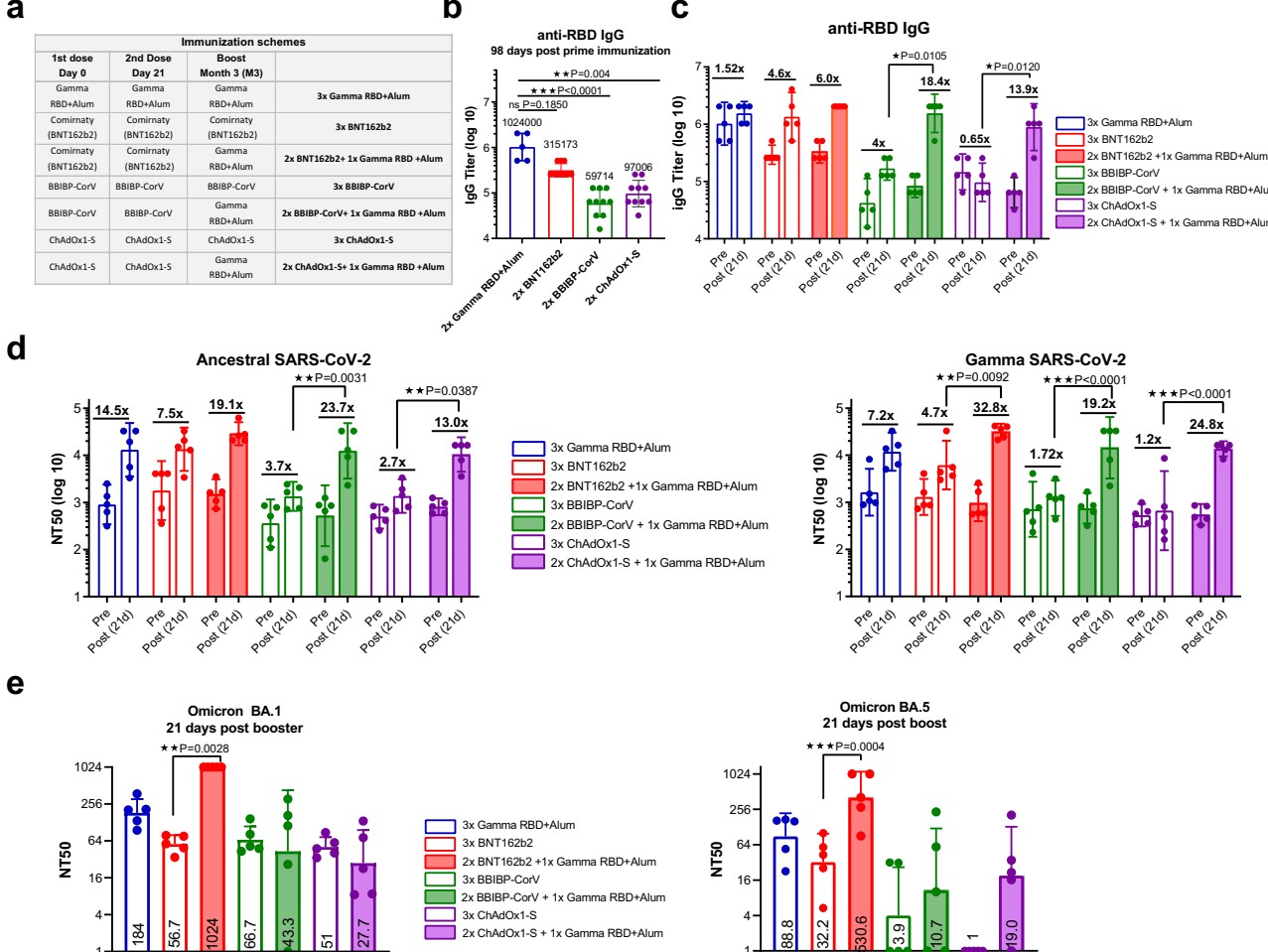

**Fig. 6 | Heterologous boosting with Gamma RBD-adapted vaccine formulation of different primary vaccine platforms. a** Immunization protocol scheme. BALB/c mice were immunized at day 0 and day 21 via i.m. with: Gamma RBD + Alum (blue), BNT162b2 (red), BBIBP-CorV (green) and ChAdOx1-S (violet). At day 100 (M3) half of mice received homologous third doses (BNT162b2, BBIBP-CorV or ChAdOx1-S accordingly to table in panel **a** and the other half of mice received heterologous booster shots with Gamma RBD + Alum. **b** Ancestral RBD-specific IgG titers in sera of immunized animals at day 98 post prime immunization. Bars are GMT ± 95% CI. Gamma RBD + Alum (n = 5 mice, blue), BNT162b2 (n = 10 mice, red), BBIBP-CorV (n = 10 mice, green) and ChAdOx1-S (n = 10 mice, violet). Kruskal Wallis test. *p < 0.05, **p < 0.01, ***p < 0.001. Exact p values are shown. **c** Comparison of RBD-specific IgG titers pre and post-boost immunization. Numbers in bold are GMT fold-increase values. Bars are GMT ± 95% CI. n = 5 mice per group. Kruskal Wallis.

*p < 0.05. Exact p values are shown. **d** Neutralizing-antibody titers against Ancestral and Gamma SARS-CoV-2 were evaluated pre (98 dpp) and post boost immunization (121 dpp) of mice. Neutralization titer was defined as the serum dilution that reduces 50% of the cytopathic effect (NT50). Bars are GMT ± SD. Numbers in bold are fold-increase of GMT values for each group. n = 5 mice per group. One-way ANOVA and Bonferroni multiple comparison test. *p < 0.05, **p < 0.01, ***p < 0.001. Exact p values are shown. **e** Neutralizing activity of antibodies against Omicron (BA.1 and BA.5) were evaluated 21 days post boost immunization of mice. Neutralization titer was defined as the serum dilution that reduces 50% the cytopathic effect (NT50). Bars are GMT ± SD and numbers are GMT values for each group of mice. n = 5 mice per group. Kruskal Wallis test. **p < 0.01, ***p < 0.001. Exact p values are shown Source data are provided as a Source Data file.

## A booster dose of Gamma RBD-based vaccine induces higher neutralizing Ab titers than homologous booster vaccination in mice previously primed with different vaccine platforms

Booster vaccination has become a strategy to prevent SARS-CoV-2 breakthrough infections. Immunogenicity of the Gamma RBD adjuvanted vaccine as a heterologous booster for different primary vaccine platforms was studied in comparison with homologous regimens (i.e., three dose of the same vaccine). To this end, each group of mice was immunized with one of the following vaccine platforms in a two-dose primary scheme vaccination: (i) non-replicating adenovirus vaccine ChAdOx1-S (Oxford AstraZeneca), (ii) the monovalent mRNA vaccine BNT162b2 (Pfizer) and (iii) the inactivated SARS-CoV-2 vaccine BBIBP-CorV (Sinopharm). Three months later, half of the animals received a homologous boost and the other half a heterologous boost with the Gamma RBD-based vaccine (Fig. 6a).

All vaccine platforms induced Ancestral RBD-specific IgG after two dose primary vaccination (Fig. 6b). Interestingly, three months after primary vaccination the Gamma RBD-based vaccine candidate elicited the highest Ab titers (day 98 post prime immunization, Fig. 6b). Three weeks after the third dose, in the group that received a primary regimen with the inactivated BBIBP-CorV- vaccine the increase in RBD-specific IgG titers was higher after heterologous boosting with Gamma-adapted vaccine than with the homologous vaccine (18.4× vs. 4×-fold-increase respectively). Similarly, primary vaccination with the non-replicating adenovirus vaccine group ChAdOx1-S and boosted with Gamma RBD showed a higher fold-increase in anti-RBD IgG titers than homologous ChAdOx1-S boosting (13.9× vs. 0.65×-fold-increase respectively, Fig. 6c). In the group that received mRNA vaccine as primary regimen the increment in RBD-specific IgG titers was similar after homologous or heterologous booster (4× vs. 6× fold rise respectively, Fig. 6c).

Analysis of the neutralizing activity of sera from mice before (pre) and after booster (post) immunization against Ancestral SARS-CoV-2 revealed that the fold-increase in neutralizing Ab titers were significantly higher after heterologous booster with the Gamma subunit vaccine than homologous booster with BBIBP-CorV and ChAdOx1-S vaccines (Fig. 6d). Heterologous boosting with the Gamma subunit vaccine induced a significantly higher increase of the neutralizing Ab titers against the Gamma VOC than homologous boosting in all vaccine platforms used to prime mice (Fig. 6d). The best neutralization activity (against Ancestral and Gamma variants) was observed in the group that received primary immunization with the mRNA vaccine BNT162b2 and boosted with the Gamma RBD-adapted vaccine (Fig. 6d).

Finally, cross-neutralization activity was assessed against Omicron subvariants BA.1 and BA.5 three weeks after the booster dose. In all groups, antibody neutralization capacity was reduced against Omicron subvariants compared to neutralizing Ab titers against the Ancestral and Gamma variant. However, a degree of cross-reactive antibodies against BA.1 and BA.5 was detected, and the groups boosted with the subunit gamma adapted vaccine outperformed groups boosted with homologous vaccines (adenoviral, inactivated and mRNA vaccines, Fig. 6d, e). In particular, in the group that received mRNA as primary vaccination, the neutralizing Ab titers against Omicron BA.1 and BA.5 were significantly higher after a heterologous boost with Gamma RBD-adapted subunit vaccine than with a homologous boost vaccination (Fig. 6d, e).

These results together indicate that the Gamma RBD-adapted recombinant vaccine candidate could serve as an effective booster of different primary vaccine platforms inducing a broad cross-reactive immunity.

## Discussion

The rapid emergence of SARS-CoV-2 variants has challenged the global development of vaccines. The ability of VOCs to escape neutralization by antibodies elicited by infection, vaccination, or therapeutic application has been widely studied[23,24]. Developing variant adapted vaccines that can elicit broad and strong immune responses against these predominant circulating variant(s) is proposed as one of the best strategies to address variants immune evasion.

In this study, we described the immunogenicity of a recombinant Gamma adapted subunit vaccine formulated with aluminum hydroxide to be used as a primary or booster vaccine. Ancestral and Gamma variant derived RBD antigens were compared in their capacity to induce nAb responses against SARS-CoV-2. The Gamma adapted adjuvanted vaccine elicited a broadly nAb response against the Ancestral virus and different VOCs compared to the formulation containing the Ancestral RBD antigen. In agreement with these results in mice, we have recently published a phase 1 study where individuals boosted with Gamma adapted RBD vaccine induced cross-reactive antibodies able to neutralize not only the Ancestral and Gamma variant but also Delta, Omicron BA.1 and BA.5 VOCs[20]. The levels of neutralizing antibodies induced in humans against Gamma, Omicron BA.1 and BA.5 VOCs were higher in comparison with those induced by a cohort of mRNA boosted individuals[20].

Gamma and Beta variants, which originated in Brazil and South Africa, respectively, carry the K417N/T, E484K and N501Y amino acid changes in the spike RBD. Mutations in the Beta variant decreased the neutralization sensitivity of convalescent and vaccinated sera by 6x-fold, while mutations in Gamma decreased neutralizing activity by 3x-fold[25].

The Gamma variant booster vaccine candidate tested here showed broader cross-reactive neutralizing antibody response against several VOCs compared to the response induced by a recombinant formulation with the Ancestral RBD or Ancestral Spike 2 P antigen indicating that antigen adaptation to a virus variant could have an important role on the immunogenicity of the formulation.

In this work, we first hypothesized and then examined possible mechanisms underlying the superior immunogenicity of the Gamma adapted vaccine. Interestingly, only the Gamma adapted vaccine increased the proportion of germinal center B cells involved in the maturation of high avidity and isotype-switched antibodies required for long-lasting protective immunity. This could be related to the notion that residue substitutions in the Gamma antigen simulate a structure that enhances the spectrum of neutralizing activity. Immunoinformatic tools used for prediction of linear B-cell epitopes in the Ancestral and Gamma RBD sequences revealed that both antigens shared one antigenic epitope, that include in the Gamma sequence the N501Y substitution. The presence of the threonine (Y) in the position 501 in the Gamma RBD sequence increased the antigenicity score of this epitope and may explain the superior immunogenicity observed with this formulation. The immunogenicity of this epitope has been previously reported and experimentally confirmed using convalescent sera from COVID-19 patients[26]. In agreement with this hypothesis and our results, a study evaluating the immunogenicity of an S-protein sequence containing the K417N, E484K and N501Y substitutions in addition to D614G (B.1) demonstrated that the variant adapted vaccine is capable of eliciting high neutralizing Ab titers that cross-reacted with five VOCs in NHP[27]. Moreover, in preclinical and clinical trials Beta variant derived vaccines induced broadly neutralizing antibodies that can cross-react against different VOC[27–29] in the same way than the Gamma derived vaccine presented in this work. Important to note, a Beta-adapted subunit vaccine has been approved for use in Europe.

A recombinant vaccine containing RBD-dimers derived from Ancestral SARS-CoV-2 and aluminum hydroxide as adjuvant proved to be safe and immunogenic in humans[30]. Structural analysis of these RBD-dimers indicates that are arranged in a way where the receptor binding motif (RBM) is correctly presented and the major antigenic sites of RBD are exposed[31]. This supports the versatility of the RBD-dimers as a module to adapt different variants for the induction of broader immune responses. Hence, it has been reported that chimeric RBD-dimers elicit broader responses to variants compared with homodimers[31]. Increasing evidence suggests that vaccine protection against COVID-19 might be waning over time, and newly emerging variants, such as Delta and Omicron, might evade vaccine-induced immune protection.

In mice, the Gamma adapted vaccine induced a long-lasting antibody response with neutralizing activity against different VOCs up to 253 days after the last immunization. High-affinity antibodies that provide durable protective immunity are produced by long-lived bone marrow plasma cells differentiated from germinal center B cells. Indeed, after immunization with the Gamma vaccine formulation total and RBD-specific LLPCs in bone marrow were observed indicating that long-term antibodies could be secreted from those cells. Adjuvanted recombinant COVID-19 vaccine formulations have reported durability of the neutralizing-antibody responses for 7 months post-immunization in NHP[32]. This study together with previous reports suggest that recombinant variant-specific vaccines may be needed for improved and long-lasting protection against known and emerging VOCs.

Main correlates of protection against COVID-19 are binding and neutralizing antibody titers[33]. However, cellular immunity is associated with positive outcomes[34]. Here, we have provided a comprehensive assessment of the T cell response to both vaccine versions. Both vaccine formulations elicited CD4+ and CD8+ T cell responses. Analysis of polyfunctionality on vaccine elicited T cells shown that the Gamma vaccine have a major proportion of T cells that produce two cytokines (IFN-γ and TNF-α). Polyfunctional Th1 responses were reported after human vaccination with adenoviral ChAdOx1-S and the mRNA BNT162b2 vaccine[35,36].

Protection studies in transgenic mice presented here confirm that the immunity induced by the Gamma-derived RBD vaccine candidate

can protect mice from a nasal challenge of Ancestral or Omicron BA.5 SARS-CoV-2 virus. Mono and bivalent candidates containing different combinations of VOCs based antigens have been assessed in humans[13,37]. Bivalent mRNA vaccines containing Omicron as boosters have been implemented in several countries. Interestingly, the Gamma adapted vaccine showed similar levels of protection than the bivalent BNT162b2 vaccine against the Omicron BA.5 variant suggesting that adaptation of monovalent or bivalent vaccines that include the Gamma variant could be a good strategy to protect against new VOCs.

Updated immunization strategies were implemented as booster shots across the globe. In this work, we showed that vaccination with Gamma RBD plus alum as booster of adenoviral, inactivated or mRNA-based vaccines induced a higher binding and neutralizing antibody titer increase against different VOCs than vaccinations with homologous vaccines. In general, heterologous boosters showed higher vaccine effectiveness than homologous boosters for all outcomes in humans, providing additional support for a mix-and-match approach[38,39]. Our vaccine not only increased GMT values of neutralizing Ab against Ancestral and Gamma virus variants but also against Omicron lineages (BA.1 and BA.5).

The Gamma-RBD vaccine candidate reported in this article has demonstrated to be safe in non-clinical and clinical safety studies. Both the antigen and the formulated vaccine are being produced under GMP at manufacturing plants of Laboratorio Pablo Cassará S.R.L. located in Buenos Aires. Stability studies showed that the vaccine can be stored and handled in refrigerated conditions (2–8 °C) and does not need freezing temperatures. The development of a vaccine that does not require stringent cold-chain transport and storage facilitates its availability at local and global supply. This Gamma RBD plus alum vaccine was named ARVAC CG and recently finished a Phase 2/3 clinical trial in which a monovalent vs bivalent version of the vaccine was evaluated (NCT05752201). Recently, monovalent Gamma and bivalent (Gamma-Omicron BA.4/5) ARVAC vaccines have been approved and authorized for use as booster in Argentina becoming the first local vaccine developed and produced in the region.

This work highlights the value of a Gamma variant-adapted vaccine to induce higher and broader immune responses against SARS-CoV-2 as primary or booster vaccination in monovalent or multivalent updated vaccines.

## Methods

### Antigen expression and purification
Constructs from Wuhan (Ancestral), Gamma and Omicron BA.4/5 SARS-CoV-2 variants of the antigen consisted of a single-chain dimer of the receptor binding domain (RBD), comprising amino acids 319 R to 537 K of the Spike protein from Ancestral and Gamma virus variant. The signal peptide sequence from the SARS-CoV-2 Spike protein was used for protein secretion. The constructs were codon-optimized for CHO cell expression and synthesized by GenScript (Hong Kong Limited, Japan). All constructs were cloned into the pJC3UMCS-4 expression vector (Pablo Cassara Foundation) comprising a CMV promoter and a cis-acting sequence for minimized host gene silencing. Linearized forms of the vectors were transfected into CHO-S cells by lipofection. After 15–20 days of G418 antibiotic selection, an end point dilution procedure was applied for clone isolation. Clone candidates were selected by specific and volumetric productivity assessed by ELISA. Recombinant protein relative abundance was confirmed by SDS-PAGE and Coomasie Brilliant Blue G-250 staining.

Pilot batches of antigen and vaccines were manufactured by Laboratorio Pablo Cassara, Argentina. Controlled cell substrates were used to inoculate a 2-L bioreactor with 1.4 L as working volume. An alternate tangential flow system (ATF, Repligen) was used for perfusion. The harvest after 20–25 days was used as starting material for a downstream process based on three chromatographic steps: first an affinity mix-mode capture chromatography step followed by two different ionic chromatography to eliminate residual host cell DNA and host cell proteins and other process-related impurities. TFF (Tangential flow filtration) was used to adjust pH, conductivity, and protein concentrations of the batches.

Recombinant SARS-CoV-2 Spike protein in its homotrimeric form, containing S1 + S2 subunits and encompassing amino acids 16-1213 was purchased in BPS biosciences (BPS Bioscience, Inc.San Diego, USA). This protein corresponds to the wild-type (strain of reference). The construct contains mutations K986P and V987P, and a T4 trimerization domain followed by a His-tag (6xHis) in C-terminal.

### SDS-PAGE and western blot analysis
RBD samples were run under reducing and non-reducing conditions by SDS-PAGE (10% polyacrylamide). The samples were visualized by staining with Coomassie Brilliant Blue G-250. Bands were then transferred to a nitrocellulose membrane (Bio-Rad Inc. Hercules, CA, USA), blocked with Non-fat milk 5% in phosphate buffered saline (PBS)-Tween 0.05%, and incubated with anti-RBD rabbit polyclonal serum (1/1000 dilution). An anti-rabbit IgG horseradish-peroxidase conjugated (1/2000 dilution) was used as a secondary antibody (Agilent DAKO, Santa Clara, CA, USA, Cat# P0448, Lot 20061231) and 4-Chloro-1-Naphthol as substrate for colorimetric detection.

### ACE2 binding to RBD analyzed by titration ELISA
ACE2 binding to the produced RBD variant antigens was analyzed by ELISA. Briefly, 0.1 ug of RBD per well was immobilized on a high binding Microplate (Greiner Bio-One GmbH, Austria.) and incubated ON at 4 °C. Next, plates were blocked with 5% non-fat milk in PBS 0.05% Tween20 for 2 h at 37 °C and then washed with PBS and 0.05% Tween20. Plates were then incubated with hACE2 (R&D, Minneapolis MN, USA) for 2 h at 37 °C and another washing step was performed. Finally, plates were incubated with anti-hACE2 (R&D Systems, Cat#: AF933. Lot: HOK0620051) diluted to a concentration of 0.5 μg/ml in 1% non-fat milk PBS 0.05% Tween 20 for 2 h at 37 °C. Detection was performed with a secondary Ab conjugated to HRP (Agilent DAKO, Santa Clara, CA, USA, Cat# P0449, Lot:41308941) at a dilution: 1/1000 in 1% non-fat milk PBS 0.05% Tween 20 for 1 h at 37 °C and visualized with tetramethylbenzidine (TMB) substrate (BD biosciences, Franklin Lakes, New Jersey, USA).

### Adjuvant and formulations
Pilot batches of vaccines were manufactured using aluminum hydroxide as adjuvant. Aluminum hydroxide 2% was supplied by CRODA (Croda International plc, United Kingdom). The antigen was adsorbed onto the adjuvant by mixing. Free antigen was controlled to be less than 10% of total antigen in the vaccine. Endotoxin levels of vaccine formulations were determined by Limulus amebocyte assay using Endosafe® Kinetic Chromogenic LAL Reagents (Charles River, Massachusetts, USA). All vaccine formulations contains <4 EU/ml.

### Ethics statement
All experimental protocols with animals were conducted in strict accordance with international ethical standards for animal experimentation (Helsinki Declaration and its amendments, Amsterdam Protocol of welfare and animal protection and National Institutes of Health, NIH USA, guidelines.). The protocols performed at UNSAM were also approved by the Institutional Committee for the use and care of experimental animals (CICUAE) from National University of San Martin (UNSAM) (Protocol number 01/2020). All procedures involving animals at Virginia Tech were approved on 04/09/2021 by the Virginia Tech's Institutional Animal Care and Use Committee (IACUC) and all animal experiments were performed in compliance with the guidelines of Virginia Tech's IACUC. Animal procedures at Universidad del Litoral were approved by the Ethics Committee of the Facultad de Ciencias Veterinarias, Universidad Nacional del Litoral, Santa Fe, Argentina

(Protocol number 808/23). Studies in vitro using SARS-CoV-2 were done in a Biosafety level 3 laboratory at IIB-UNSAM, and the protocol was approved by the IIB Institutional Biosafety Committee.

## Animals and immunizations in immunogenicity studies

Eight-week-old female BALB/c or C57BL/6 mice were obtained from IIB-UNSAM animal facility. For immunogenicity studies, animals were intramuscularly (i.m) inoculated at day 0 and 14 with (i) Gamma RBD (10 μg) + Alum (100 μg), (ii) Ancestral RBD (10 μg) + Alum (100 μg) or (iii) placebo (Alum alone). In other experiments, BALB/c mice were vaccinated with: (i) Gamma RBD (10 μg) + Alum (100 μg) or (ii) placebo. A group containing ancestral spike 2 P (5 μg, BPS Biosciences) and Alum (100 μg) was used as control in some experiments. Blood samples were collected weekly to measure total and neutralizing antibody titers and at day 42 post prime immunization animals were sacrificed and spleens harvested.

In booster experiments, eight-week-old female BALB/c mice were i.m inoculated on days 0 and 21 with (i) Gamma RBD (10 μg/dose) + Alum (100 μg/dose, $n = 5$ mice), (ii) Comirnaty (1 μg/dose) (Pfizer-BioNTech, $n = 10$ mice), (iii) BBIBP-CorV (1.3 U/dose containing 45 μg of Alum) (Sinopharm, $n = 10$ mice), (iv) and ChAdOx1 nCoV-19 ($3 \times 10^9$ particles/dose) (Oxford-AstraZeneca, $n = 10$ mice). At day 99, 5 animals per group were boosted with the Gamma RBD (10 μg) + Alum (100 μg) while the other 5 animals received the homologous vaccine. Blood samples were collected weekly to measure total and neutralizing antibody titers.

## Determination of antibody levels in serum

RBD-specific antibody responses (IgG) were evaluated by indirect ELISA as described previously[40]. Briefly, plates were coated with 0.1 μg/well of RBD (derived from Ancestral, Gamma or Omicron BA.4/5 variant) in phosphate buffered saline (PBS) overnight at 4 °C. Plates were washed and incubated with diluted sera and then plates were washed and incubated with a polyclonal HRP conjugated anti-mouse IgG (SIGMA, St. Louis, MO, USA, Cat #A4416-1ML) at a dilution of 1/2000. Results were read at 450 nm to collect end point ELISA data. End-point cut-off values for serum titer determination were calculated as the mean specific optical density (OD) plus 3 standard deviations (SD) from sera of saline immunized mice and titers were established as the reciprocal of the last dilution yielding an OD higher than the cut-off.

## Viruses

Ancestral SARS-CoV-2 (B.1, GISAID Accession ID EPI_ISL_16290469), Alpha (B.1.1.7 GISAID Accession ID EPI_ISL_15806335), Gamma (P.1, GISAID Accession ID EPI_ISL_15807444), Omicron BA.5 (GISAID Accession ID EPI_ISL_16297058) were isolated from nasopharyngeal specimens at the Instituto de Investigaciones Biotecnológicas (IIB-UNSAM) and adapted in Vero E6 cultures. Delta SARS-CoV-2 (GISAID Accession ID: EPI_ISL_11014871) and Omicron BA.1 (GISAID Accession ID EPI_ISL_10633761) were isolated at Instituto de Investigaciones Biomédicas en Retrovirus y SIDA (INBIRS, UBA-CONICET) from nasopharyngeal swabs of patients.

## Determination of Ag-Specific B Cells and T follicular helper cells

Ag-specific B cells (plasmablasts and germinal center B cells) present in the spleens were determined by flow cytometry. For B cells and plasmablasts, splenocytes were plated ($2 \times 10^6$ cells/well) and stained with Viability dye (Zombie Acqua, Biolegend), anti-B220 Alexa Fluor 488 (Biolegend, Cat #103225, clone RA3-6B2, 1/400), anti-CD19 APC/Cy7 (Biolegend, Cat #103225, clone RA3-6B2, 1/400), anti-CD138 Brilliant Violet 785 (Biolegend, Cat #142534, clone 281-2, 1/200), anti-IgD Brilliant Violet 605 (Biolegend, Cat #405727, clone 11-26c2a, 1/150), anti-IgG PECy7 (Biolegend, Cat #405315, clone Poly4053, 1/200). For Germinal center cells splenocytes were stained with Viability dye

(Zombie Acqua, Biolegend) anti-B220 Alexa Fluor 594 (Biolegend, Cat #103254, clone RA3-6B2, 1/100), anti-CD19 APC/Cy7 (Biolegend, Cat #103225, clone RA3-6B2, 1/400), anti-IgD Brilliant Violet 605 (Biolegend, Cat #405727, clone 11-26c2a, 1/150), anti-GL7 Alexa Fluor 488 (Biolegend, Cat #144612, clone GL7, 1/200)and anti-CD95 PE Biolegend, Cat #152608, clone SA36758, 1/800). Detection of LLPCs was performed by staining with stained with Viability dye (Zombie Acqua, Biolegend) anti-B220 Alexa Fluor 488 and anti-CD138 Brilliant Violet 785. For Ag-specific detection, cells were also stained with fluorescent Gamma or Ancestral RBD (in-house RBD conjugated to Alexa Fluor 647 succinimidil ester). Next, cells were washed, fixed, and analyzed by flow cytometry. T follicular helper cells were evaluated by flow cytometry in spleen. Briefly, $2 \times 10^6$ cells/well were plated and stained with Viability dye (Zombie Acqua), anti-CD4 Brilliant Violet 711 (Biolegend, Cat #100447, clone GK1.5, 1/400), anti-CD8 Alexa Fluor 700, anti-CXCR5 PeCy7 (Biolegend, Cat #100730, clone 53-6.7, 1/200) and anti-PD-1 Brilliant Violet 421 (Biolegend, Cat #135221, clone 29 F.1A12, 1/100). Next, cells were washed, fixed, and analyzed by flow cytometry (BD Fortessa LSR X-20, BD Biosciences).

## SARS-CoV-2 neutralization assay

Serum samples were heat-inactivated at 56 °C for 30 min. Serial dilutions were performed and then incubated for 1 h at 37 °C in the presence of 300 $TCID_{50}$ of SARS-CoV-2 in DMEM 2% FBS. One hundred μl of the mixtures were then added onto Vero cells monolayers. After 72 h at 37 °C and 5% $CO_2$, cells were fixed with PFA 4% (4 °C overnight) and stained with crystal violet solution in methanol. The cytopathic effect (CPE) of the virus on the cell monolayer was assessed by surface scanning at 585 nm in a microplate reader (FilterMAx F5 Microplate reader, Molecular Devices, San Jose, CA, USA). Average optical density (OD) of each well was used for the calculation of the percentage of neutralization of viral CPE for each sample as: Neutralization $\% = 100 \times (DO_{sample} - DO_{virus\ control}) / (DO_{cell\ control} - DO_{virus\ control})$. Non-linear curves of Neutralization (%) vs. Log 1/sera dilution were fitted to obtain the titer corresponding to the 50% of neutralization (NT50).

## Antigenic cartography

The R package Racmacs (https://acorg.github.io/Racmacs/index.html) was used to create antigen cartography maps from serum neutralization titers against the SARS-CoV-2 live viruses (Alpha, Beta, Gamma, Delta, Omicron BA.1 and Omicron BA.5 VOC and the original strain Wuhan). Antigenic distances are measured in antigenic units (AU). One AU corresponds to a two-fold dilution of the antibody in the neutralization assay. Each square in the map indicates 1 AU. The antigenic distance is measured in any direction of the map.

## Determination of T cell immune responses

Four weeks after the second dose, mice were sacrificed to study cellular responses. Splenocytes were cultured for 5 days in the presence of RBD antigen (Ancestral or Gamma) or complete medium. Then, supernatants were collected, and cytokines were measured by flow cytometry with the multiplex bead assay LEGENDplex™ MU Th Cytokine Panel (12-plex) (Biolegend. San Diego, CA. Cat# 741043). For intracellular cytokine determination: splenocytes were cultured ($4 \times 10^6$ cells/well) in the presence of stimulus medium (complete medium supplemented with anti-CD28 -Biolegend,.Cat #103710, clone 9C10, 1/1000- and anti-CD49d -Biolegend Cat #103710, clone 9C10, 1/1000-) or Ag stimuli (stimulus medium + 2 ug/ml of Ancestral or Gamma RBD-peptides (JPT Peptide Technologies GmbH, Berlin) + 10 ug/ml of Ancestral or Gamma RBD proteins) for 18 h. RBD peptides comprises a pool of 53 peptides derived from a peptide scan (15mers with 11 aa overlap) through Receptor Binding Domain of Spike Glycoprotein (Swiss-Prot ID: P0DTC2, region: 319–541) of SARS-CoV-2 from the Wuhan or Gamma variant (Lineage P.1, B.1.1.28.1). Next,

brefeldin A was added for 5 h to the samples. Cells were then washed, fixed, permeabilized, stained, and analyzed by flow cytometry. The cells were stained with Viability dye (Zombie Acqua), anti-mouse-CD8a Alexa Fluor 488 (Biolegend, Cat #100723, clone 53–6.7, 1/100), anti-mouse-CD4 Alexa Fluor 647 (Biolegend, Cat #100424, clone 6K1.5, 1/200), anti-TNFα PeCy7 (Biolegend, Cat #506324, clone MP6-XT22, 1/50)., anti-IL-2 BV 421 (Biolegend, Cat #503826, clone JES6-5H4, 1/50) and anti-IFN-γ P (Biolegend, Cat #505808, clone XMG1.2, 1/50). BD Fortessa LSR-X20 cytometer with DIVA Software were used for analysis.

### SARS-CoV-2 challenge of vaccinated K18-hACE2 mice

For the Omicron BA.5 challenge experiment, transgenic B6.Cg-Tg(K18-ACE2)2Prlmn/J (K18-hACE2, JAX stock #034860) were purchased from Jackson Laboratories (Bar Harbor, Maine, USA), transported to Argentina and housed and expanded at the enforced BSL2+ Facility of the Center for Comparative Medicine (ICIVET-Litoral, UNL-CONICET).

Four-month-old K18-hACE2 mice ($n = 12$/per group, 6 males and 6 females per group) were i.m inoculated at day 0 and 14 with 1/5 of human doses of: (i) Gamma RBD (10 μg) + alum (100 μg), (ii) Ancestral RBD (10 μg) + alum (100 μg), (iii) bivalent BNT162b2 (6 ug) or (iv) placebo (Alum). For virus challenge, at day 28 post prime immunization, mice were anesthetized (ketamine/xylazine) and infected intranasally with 20 μl containing $1 \times 10^4$ PFU of Omicron BA.5 (GISAID Accession ID EPI_ISL_16297058) split evenly in each nare. Clinical signs of disease (weight loss, rapid breathing, hunched posture and inactivity) were monitored daily and a half of the animals per gender per group were euthanized at day 3 and 5 post-infection. Lungs were obtained, the left half was homogenized and used for viral RNA determination and right half were used to histopathological analyses. A control untreated and uninfected group of mice were sacrificed to characterize background lesions.

For the challenge with the Ancestral virus (WA1/2020 strain), four-week-old K18-hACE2 mice from Jackson Laboratory were used. Animals were housed at the Department of Entomology, College of Agriculture and Life Sciences, Fralin Life Science Institute, Virginia Polytechnic Institute. Mice were separated into two groups: (i) control ($n = 8$ mice) Placebo (PBS) and (ii) Gamma RBD (10 μg) + Alum (100 μg) ($n = 9$ mice). Mice in each group included males and females. They were i.m immunized at day 0 and 14 with the same doses described for immunogenicity assays. Four weeks post second vaccination, mice were challenged intranasally with $10^5$ PFU of SARS-CoV-2 strain WA1/2020 split evenly in each nare. Mice were monitored daily for weight loss and signs of disease for two weeks post-challenge. Three mice per group were euthanized on day 5 post-challenge to evaluate organ viral loads, by plaque assay on Vero E6 cells as previously described[40].

### Determination of viral RNA by RT-qPCR

Tissues were weighed and homogenized in DMEM media supplemented with 2% heat-inactivated FBS. Tissue homogenates were clarified by centrifugation at $20,000 \times g$. for 5 min and stored at −80 °C. RNA was extracted from 10% organ homogenates using Trizol as lysis reagent (TRIzol®, Thermo Fisher Scientific) and following manufacturing instructions. PCR was performed using the WHO recommended method as the reference technique to detect the RdRp gene, as previously described[41]. Oligonucleotides: RdRp_SARSr-F (sequence GTGARATGGTCATGTGTGGCGG), RdRp_SARSr-R (sequence CARATGTTAAASACACTATTAGCATA), RdRp_SARSr-P2 (sequence FAM-CAGGTGGAACCTCATCAGGAGATGC-BBQ) and RdRP_SARSr-P1 (sequence FAM-CCAGGTGGWACRTCATCMGGTGATGC-BBQ). The standard curve was generated using a viral RNA secondary standard calibrated from the international standard SARS-like Wuhan ivRNA E, RdRp and N Genes; $1 \times 10^8$ copies/uL provided by PAHO-WHO.

### Histopathological studies

Tissues collected at necropsy were fixed in 4% buffered formaldehyde for 8–10 h at room temperature and then washed in PBS. Later, fixed tissues were dehydrated in an ascending series of ethanol, cleared in xylene and embedded in paraffin. Sections (4 μm thick), obtained by rotative microtome, were mounted on slides treated previously with 2% (v/v) 3-aminopropyltriethoxysilane (Sigma-Aldrich, Saint Louis, MO, USA) and stained with hematoxylin-eosin for the histopathology analysis.

Sections were examined by a qualified veterinary pathologist who was blinded to the animal and treatment groups. A score of 0 to 3 based on absent, mild, moderate, or severe degree was used to describe the pulmonary pathology manifestations such as: type II pneumocyte hyperplasia, hemorrhages, edema and leucocytes infiltration. A cumulative pathology score also was obtained for each animal.

### Statistical analysis

Statistical analysis and plotting were performed using GraphPad Prism 8 software (GraphPad Software, San Diego, CA). In experiments with more than two groups, data were analyzed using one-way ANOVA with Kruskal Wallis or Bonferroni post-test. When necessary, a logarithmic transformation was applied prior to the analysis to obtain data with a normal distribution. In experiments with two groups, an unpaired $t$ test or a two-sided Mann–Whitney $U$ test were used. A $p$ value < 0.05 was considered significant. When exact $P$ values are not shown we used the following reference: *$P < 0.05$ **$P < 0.01$ and ***$P < 0.001$. When bars were plotted, results were expressed as means ± SEM or GMT ± SD for each group. Survival curves were analyzed using Long-rank test.

### Reporting summary

Further information on research design is available in the Nature Portfolio Reporting Summary linked to this article.

## Data availability

All data supporting the findings of this study are available within the paper. Source Data are provided with this paper. Any additional information related to the study is available from the corresponding author upon reasonable request. Source data are provided with this paper.

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

## Acknowledgements

The authors are grateful to the technical staff at the I + D Biofármacos Laboratorio Pablo Cassará and to Fundación Pablo Cassará for their significant assistance in the preparation and control of the antigen pilot lots and the regulatory documentation: Cintia Parsza, Brenda Heinrich, Melisa Gambone, Horacio Descoins, Christian Cortez, María Victoria Román and Romina Albarracín. The authors would also like to thank Danielle Porier, Krisangel López, and Manette Tanelus for technical assistance with the SARS-CoV-2 challenge studies. We are grateful to the technician support of CMC (ICiVet-Litoral) staff. We thank Ministerio de Salud de la Provincia de Buenos Aires for providing mRNA, adenoviral and inactivated vaccines. This work was supported by Laboratorio Pablo Cassará and grants from Agencia Nacional de Promoción de la Investigación, el Desarrollo Tecnológico y la Innovación (AGENCIA I + D + i) and Ministerio de Ciencia, Tecnología e Innovación (FONARSEC 0001) to J.C. and from the National Institute of Allergy and Infectious Diseases of the National Institutes of Health under Award Number R01AI153433 to A.J.A.

## Author contributions

L.M.C. was responsible of overall experiment design, conducted humoral and cellular experiments, collected data, performed data analysis, and wrote the manuscript. J.C. was responsible for overall experimental design and conceptualization, wrote the manuscript and received funding support. J.M.R. generated high expression vectors, designed the antigen and was responsible of development of the vaccine formulation and downstream development. Coordination of antigen and vaccine production. K.A.P. and J.M.F. participated in experimental design, data analysis and reviewed the manuscript. D.E.A. designed and supervised neutralization studies and analysis. A.D. conducted and analyzed humoral studies and conducted cellular studies. C.P.C., L.A.B, L.M.S and L.P. conducted humoral and cellular studies. M.R.M. and E.F.C. performed virus neutralization studies and data analysis. S.A.DP. generated high expression vector and optimization of high

productive clones. A.C.HI. Responsible of upstream and downstream development of antigens. I.G.K. coordinated overall analytics procedures, developed validation and analytics methodologies, performed structural characterization of antigen and vaccine formulation. A.V.DN. generated cell banks, performed stability, clones purity and identity studies. V.K. and I.D. generated, characterized, and screened cell clones. F.M.Z., J.A.B. and C.J.P performed upstream and downstream development of antigen. J.E.S. and M.LC. participated in the development of validation and analytics methodologies, performed structural characterization of antigen and vaccine formulation and performed animal studies. J.C.V. participated in the development of vaccine formulation, collaborates in the overall process of vaccine development. A.J.A. and A.N.A design, supervise and conduct animal challenge studies and data analysis. H.O. and A.C. collaborate in the challenge experiment design and preparation of viral stocks. W.B.S., U.S.N., P.U.D, and T.S. performed animal challenge studies and histopathology data analysis. M.A performed RT-PCR to determine viral loads. All authors revised the manuscript.

## Competing interests

J.M.R., A.C.H.I., F.M.Z. are salaried employees of Fundacion Pablo Cassara. S.A.D.P., I.G.K., A.V.D.N., V.K., I.D., C.J.P., J.A.B., J.E.S., M.L.C., J.M.F. and J.C.V. are salaried employees of Laboratorio Pablo Cassara. L.M.C., J.C., K.A.P., D.E.A., A.D., C.P.C., L.A.B, L.M.S., L.P., M.R.M., E.F.C. A.J.A., W.B.S., A.N.A., U.S.N., P.U.D., M.A., A.C. and T.S. declare no competing interests relevant to this article.

## Additional information

¹Instituto de Investigaciones Biotecnológicas, Universidad Nacional de San Martín (UNSAM) – Consejo Nacional de Investigaciones Científicas y Técnicas (CONICET), San Martín (1650), Buenos Aires, Argentina. ²Escuela de Bio y Nanotecnologías (EByN), Universidad Nacional de San Martín, San Martín (1650), Buenos Aires, Argentina. ³Laboratorio Pablo Cassará, Unidad de I+D de Biofármacos, Ciudad Autónoma de Buenos Aires, Buenos Aires C1440FFX, Argentina. ⁴Fundación Pablo Cassará, Unidad de I+D de Biofármacos, Ciudad Autónoma de Buenos Aires, Buenos Aires C1440FFX, Argentina. ⁵Department of Entomology, College of Agriculture and Life Sciences, Fralin Life Science Institute, Virginia Polytechnic Institute and State University, Blacksburg, VA 24061, USA. ⁶Centro de Medicina Comparada, ICiVet-Litoral, Universidad Nacional del Litoral-CONICET; Esperanza, Santa Fe 3080, Argentina. ⁷Servicio Virosis Respiratorias, Laboratorio de Referencia de Influenza, SARS-CoV-2 y otros Virus Respiratorios, Centro Nacional de Influenza de OPS/OMS, Departamento de Virología, Instituto Nacional de Enfermedades Infecciosas - ANLIS "Dr. Carlos G. Malbrán". Ciudad Autónoma de Buenos Aires, Buenos Aires C1282AFF, Argentina. ⁸Facultad de Medicina UBA, Instituto de Investigaciones Biomédicas en Retrovirus y SIDA, INBIRS-CONICET, Buenos Aires, Argentina. ⁹Laboratorio Pablo Cassará - I+D+i, Ciudad Autónoma de Buenos Aires, Buenos Aires C1408GBV, Argentina. ¹⁰Center for Emerging, Zoonotic, and Arthropod-borne Pathogens, Virginia Polytechnic Institute and State University, Blacksburg, VA 24061, USA. ✉e-mail: lcoria@iib.unsam.edu.ar; jucassataro@iib.unsam.edu.ar

