## [Peer Review File · Nature Communications]

A Gamma-adapted subunit vaccine induces broadly neutralizing antibodies against SARS-CoV-2 variants and protects mice from infectionReviewers' Comments:

Reviewer #1:

Remarks to the Author:

This study led by Coria et al. described the production of a RBD subunit vaccine derived from the Gamma variant and showed that this vaccine somehow is more immunogenic than the RBD of the ancestral variant and induced a broad-spectrum antibody against many new variants of concern, including those in the Omicron lineage. While some of the findings are intriguing and worth further exploration, I felt the conclusion of this study is a bit of an overstatement for the following reasons:

1) The Gamma variant derived RBD may be slightly more immunogenic than the RBD of the ancestral variant. However, given the currently authorized vaccines all contain both S1 and S2 with the 2-P mutations that lock the S protein in a pre-fusion conformation, the authors should have designed experiments to compare their vaccine with the 2-P S protein to demonstrate novelty of the new design. Otherwise, it is unclear how the authors' vaccine design will improve over existing vaccine design. The enthusiasm for the study has been severely dampened because of lacking such information.

2) Several studies could have been characterized in greater details. For example, in Figure 3 the fold of reduction in antibody titers against BA.5 is nearly on par with what is being reported in the literatures from mRNA vaccinated individuals. Figure 5: which strain/variant of SARS-CoV-2 was used in this challenge study? There were no histopathology analyses of animal tissues. Vaccinated animals still had 30-40% fatality, implying a not-so-impressive protection. Overall, the authors conducted a preliminary analysis, but some more comprehensive analyses will really bring the study to the next level.

The manuscript will also benefit from some language service. Examples include but not limited to:

Line 43: "One strategy to prevent breakthrough infections is developing safe and effective broad-spectrum vaccines". Not sure preventing breakthrough infections should be the goal of developing a vaccine as there is no evidence that vaccines can prevent breakthrough infections.

Line 49-51: please fix the grammar of "Vaccine-induced immunity protected K18-hACE2 mice from intranasal challenge with SARS-CoV-2 increasing survival, reducing body weight loss and viral burden in the lungs and brain".

Line 52-55. Please consider rewriting this sentence to smooth out the language.

Line 77: Wanning should be waning.

Reviewer #2:

Remarks to the Author:

The paper by Coria et al. investigates a Gamma-adapted SARS-CoV-2 subunit vaccine. RBD from the Gamma variant was formulated in alum and compared with RBD from the ancestral Wuhan-Hu-1 strain. The Gamma-adapted vaccine was immunogenic and elicited B and T cell responses in BALB/c mice and partially protected against SARS-CoV-2 challenge in the K18-ACE2 mouse model. It is not clear whether protection was against a Gamma-adapted strain. The vaccine also demonstrated a potent effect as heterologous booster of different vaccine platforms in mice. Surprisingly, compared to the ancestral RBD, the Gamma-adapted RBD was more immunogenic (in BALB/c mice).

General critique:

It is highly surprising that the Gamma-adapted RBD vaccine elicited higher IgG responses than

ancestral RBD, both against Gamma RBD and ancestral (Wuhan-Hu-1) RBD. Whilst this may be interesting, the authors do not follow up on these findings. Throughout the manuscript, details on study design, number of repeats and representative stainings are lacking.

Specific comments:

- In figure 2C, the Gamma-adapted RBD vaccine elicited higher IgG responses than ancestral RBD, both against Gamma RBD and ancestral RBD. Why was the Gamma-adapted RBD more immunogenic than ancestral RBD? Could this be due to impurities in the antigen preparations? Did the authors test for pyrogens? The experiment was apparently only performed once. A surprising finding like this requires repeat studies.
- Similarly, in figure 2E, the Gamma-adapted RBD vaccine elicited higher CD4+ T cell responses (in BALB/c mice) than ancestral RBD. What could be the reason? Do mutations in the Gamma-adapted RBD give rise to novel immunodominant CD4 T cell epitopes? In this case, it is highly possible that the finding is species-specific or even mouse strain-specific. This should at least be discussed.
- Figure 2E: Representative flow cytometry plots are missing.
- Figure 5: What virus strain was used for challenge? Is it a Gamma virus?
- Figure 5: Whilst lung virus titers appear to be low in vaccinated animals, survival was not 100%. Did remaining mice have signs of lung pathology?
- Figure 6: Experimental details are lacking for immunization studies with the different platforms.
- Figure 6B-C: Were responses measured against Gamma-specific RBD?
- What was the antigen dose and route of immunization?
- What was the dose of alum used for immunization?
- The supplier of Alum is not stated

Reviewer #3:

Remarks to the Author:

This manuscript by Coria et al. reports on an alum-adjuvanted Gamma-adapted recombinant subunit vaccine as a broad-spectrum vaccine candidate. The authors compared antibody responses in mice stimulated with alum-adjuvanted ancestral and Gamma RBD antigens, showing the gamma-adapted recombinant subunit vaccine has a potential to elicit neutralizing antibodies against different SARS-CoV-2 variants. The authors also studied the cellular immune responses stimulated by gamma-adapted recombinant subunit vaccine. Moreover, the authors demonstrated vaccine-induced protection of K18-hACE2 mice from intranasal viral challenge. This gamma-adapted recombinant subunit vaccine is also shown to enhance antibody responses as a heterologous booster for different vaccine platforms. Broadly neutralizing antibodies elicited by this gamma-adapted recombinant subunit vaccine are highly clinically relevant. However, the mechanism of action for this enhanced antibody breadth is not defined. It would add significantly to the paper if the authors were able to obtain any further insights into how this vaccine achieves this effect. In addition, the manuscript has several notable weaknesses that need to be further addressed.

1. In Fig 2C, the authors showed that alum-adjuvanted gamma RBD improved anti-Gamma IgG and anti-ancestral IgG more than alum-adjuvanted ancestral RBD. However, it is unclear that whether alum-adjuvanted gamma RBD outperforms alum-adjuvanted ancestral RBD against other SARS-CoV-2 variants. In Fig 3B, the authors showed neutralizing antibodies elicited by alum-adjuvanted gamma RBD. To make an apple-to-apple comparison, the authors should also include the alum-adjuvanted ancestral RBD in this figure.
2. Germinal center responses and follicular T responses are often linked to antibody responses. To gain a better understanding of the underlying mechanisms for vaccine-induced cross-reactive antibody responses, the author should investigate these humoral immune responses in both draining lymph nodes and spleens after immunization.

3. Fig 2E shows the cellular immune responses elicited by alum-adjuvanted gamma RBD and alum-adjuvanted ancestral RBD. It's unclear if there's any statistical significance between the two vaccines in promoting T-cell responses. Specifically, there doesn't seem to be any difference in CD8 T cell responses between the two vaccines.
4. In Fig 3A, the authors showed long-lasting antibody responses elicited by alum-adjuvanted gamma RBD. Does it correlate with plasma cells accumulating in the bone marrow?
5. The authors characterized IL-5 cytokine in Fig 4A. The authors should also characterize other crucial Th2 cytokines including IL-4, IL-10, IL-13, and IL-21 as they are often highly correlated with vaccine-induced humoral immune responses and antibody responses.
6. The label in the right panel of Fig 4B is not correct. It should be "CD8 T cell responses in spleen".
7. To make an apple-to-apple comparison, the appropriate control in Fig 4 should be alum-adjuvanted ancestral RBD instead of a placebo.
8. To demonstrate the benefits of broad antibody responses elicited by alum-adjuvanted gamma RBD vaccine, the author should challenge mice with a different strain of the SARS-CoV-2 virus. Also, the appropriate control in this figure should be alum-adjuvanted ancestral RBD instead of a placebo.

Point-by-point response to the reviewers' comments

We would like to thank the reviewers for the valuable comments, which in our opinion have further improved our manuscript.

Please find below a point-by-point response to the reviewers' comments. Reviewers' comments are written in black and responses in blue.

Reviewer #1 (Remarks to the Author):

This study led by Coria et al. described the production of a RBD subunit vaccine derived from the Gamma variant and showed that this vaccine somehow is more immunogenic than the RBD of the Ancestral variant and induced a broad-spectrum antibody against many new variants of concern, including those in the Omicron lineage. While some of the findings are intriguing and worth further exploration, I felt the conclusion of this study is a bit of an overstatement for the following reasons:

1) The Gamma variant derived RBD may be slightly more immunogenic than the RBD of the Ancestral variant. However, given the currently authorized vaccines all contain both S1 and S2 with the 2-P mutations that lock the S protein in a pre-fusion conformation, the authors should have designed experiments to compare their vaccine with the 2-P S protein to demonstrate novelty of the new design. Otherwise, it is unclear how the authors' vaccine design will improve over existing vaccine design. The enthusiasm for the study has been severely dampened because of lacking such information.

We appreciate the reviewer's suggestion to compare our results with spike as an antigen. Comparison between the antibody response induced by two doses of the BNT162b2 mRNA vaccine from Pfizer (Ancestral spike 2P) in parallel with the responses induced by Gamma RBD+Alum was shown in the original version of the manuscript (Figure 6b). In this figure, we can observe higher IgG titers against Ancestral RBD after immunization with the recombinant Gamma derived RBD vaccine in comparison to the parental BNT162b2 vaccine (spike 2P). In addition, a booster dose of the Gamma RBD formulation outperformed a booster dose of BNT162b2 in BNT162b2 previously vaccinated animals in terms of the nAb titers against Omicron BA.1 and BA.5 VOCs (Figure 6e).

Besides and to accurately answer the reviewer's question we performed new experiments to compare immune responses between the Gamma RBD vaccine and recombinant protein Ancestral spike 2P with Alum.

Thus, mice were immunized on days 0 and 14 with i) recombinant Ancestral Spike protein containing the mutations K986P and V978P (Spike 2P) plus Alum, ii) Gamma RBD plus Alum or iii) Ancestral RBD plus Alum.

Immunization with the Gamma RBD vaccine elicited higher titers of specific IgG antibodies against Ancestral, Gamma and Omicron BA.4/5 RBD antigen than administration with Spike 2P plus Alum- or the Ancestral RBD plus alum formulations. Also, a broader antibody response against different VOCs was observed with the Gamma RBD antigen in comparison with spike 2P or RBD Ancestral antigens. These new results together with previous data obtained with the Ancestral RBD suggest that it is not the Ag design of the vaccine, but the variant selected that has an important role in the immunogenicity of this formulation. We added this data in Supplementary Figure 1 of the revised version of the manuscript and references to this figure were added in results and discussion sections.

“Besides, vaccine formulation containing Gamma RBD also induced a higher antibody response with broader neutralizing activity than immunization with a formulation containing recombinant Ancestral Spike 2P as antigen plus Alum (Supplementary Fig. 1).” (Lines 139-142. Results section).

“The Gamma variant booster vaccine candidate tested here showed broader cross-reactive neutralizing antibody response against several VOCs compared to the response induced by a recombinant formulation with the Ancestral RBD or Ancestral Spike 2P antigen indicating that antigen adaptation to a virus variant could have an important role on the immunogenicity of the formulation.” (Lines 365-369. Discussion section).

Regarding novelty of the vaccine design, the recombinant vaccine tested here comprises an adjuvanted dimeric version of Gamma variant RBD protein that, to our knowledge, has not been proven in animals before. While there have been described in other experimental adjuvanted recombinant vaccines based on RBD (monomeric or dimeric form), we found that the Gamma version of RBD is more immunogenic than Ancestral RBD antigen and can induce a broader immune response against different VOCs.

2) Several studies could have been characterized in greater details. For example, in Figure 3 the fold of reduction in antibody titers against BA.5 is nearly on par with what is being reported in the literatures from mRNA vaccinated individuals.

In Figure 3 of the original version, we showed neutralizing antibodies (nAbs) titers against BA.5 variant at long-term time points (253 days post prime immunization). In the revised version of the manuscript, we added a new graph (Figure 3a) comparing the nAbs titers between the Gamma and Ancestral version of the RBD vaccine at day 42 post prime immunization. The fold reduction in nAbs titers between the Ancestral virus (Wuhan) and Omicron BA.5 was 6.4x at 42dpp and 7.7x at 253dpp. Regarding the reviewer’s comment, we think that it is difficult to make a comparison between the neutralizing antibody responses in mice and humans, even more against a variant like BA.5 that appeared in 2021 after circulation of other variants and administration of vaccines.

Unpublished data from our group in mice immunized with two doses of the monovalent parental mRNA (BNT162b2) or Gamma RBD+Alum showed that 83 days after the second dose, the fold reduction of nAbs titers against Ancestral and BA.5 VOC was 9.9x for the recombinant vaccine and 43.5x for mRNA vaccinated mice (see graph below).

Figure. BALB/c mice were immunized on days 0 and 14 with: Gamma RBD+alum or BNT162b2 (in a volume that corresponds to 1/5 of the human dose). Neutralizing antibody titers against Ancestral, Gamma and Omicron BA.1 and BA.5 variants were determined at day 83 post second dose. Fold reduction values compared to neutralization titers against Ancestral virus are shown above the bar for each group. Points are data from each individual mouse. Bars are GMT±SD. GMT values are written inside each bar.

Moreover, we have recently published the results of the phase 1 clinical study of the Gamma RBD adjuvanted vaccine done in early 2022 (Pasquevich, Coria et al. 2023) in this work we showed that boosting with the Gamma adapted vaccine induced broad nAbs that cross-react with different VOCs including Omicron BA.5 (live virus assay). In this study, the fold reduction of nAb titers between Ancestral and BA.5 variant was 8.3x in a cohort of 20 individuals (50µg RBD/vaccine dose). In addition, we had access to samples from 21 individuals from our country that received a booster dose of BNT162b2 in early 2022. We performed analysis of these samples in parallel to the samples of the phase 1 study. It was observed that mRNA boosted individuals presented a fold reduction of 29.6x in the nAbs titers against Omicron BA.5 SARS-CoV-2 compared to those against the Ancestral SARS-CoV-2.

Thus, we can confirm that results obtained in this work in mice are similar to results observed in vaccinated individuals and in both cases the Gamma recombinant adapted vaccine induced a broadly neutralizing antibody response against Ancestral, Gamma, Delta, Omicron BA.1 and BA.5.

We added a phrase in the discussion section about comparison of neutralizing antibodies against Omicron lineages in mice and humans with the Gamma RBD vaccine.

“The Gamma adapted adjuvanted vaccine elicited a broadly nAb response against the Ancestral virus and different VOCs compared to the formulation containing the Ancestral RBD antigen. In agreement with these results in mice, we have recently published a phase 1 study where individuals boosted with Gamma adapted RBD vaccine induced cross-reactive antibodies able to neutralize not only the Ancestral and Gamma variant but also Delta, Omicron BA.1 and BA.5 VOC (Pasquevich, Coria et al. 2023). The levels of neutralizing antibodies induced in humans against Gamma, Omicron BA.1 and BA.5 VOCs were higher in comparison with those induced by a cohort of mRNA boosted individuals (Pasquevich, Coria et al. 2023).” (Lines 351-359. Discussion section).

Figure 5: which strain/variant of SARS-CoV-2 was used in this challenge study? There were no histopathology analyses of animal tissues. Vaccinated animals still had 30-40% fatality, implying a not-so-impressive protection. Overall, the authors conducted a preliminary analysis, but some more comprehensive analyses will really bring the study to the next level.

In the challenge experiment of Figure 5 (original manuscript) we used the Ancestral/Wuhan virus strain. This was specified in the methods section: *“mice were challenged intranasally (i.n.) with 10⁵ PFU of SARS-CoV-2 strain WA1/2020”* (Line 639).

As suggested by the reviewer we added in the Supplementary Figure 7 data from histopathology analyses of lung tissues. Pathological score parameters of i) alveolar cellularity, ii) alveolar macrophages, iii) edema and iv) type 2 pneumocytes hyperplasia were plotted for each mouse of placebo and vaccinated groups in Supplementary Figure 7d. Representative images of placebo and immunized mice were added in Supplementary Figure 7e.

Reviewer 3 suggested we include a challenge experiment with another VOC. Thus, we performed an Omicron BA.5 SARS-CoV-2 challenge experiment in animals vaccinated with i) the Gamma RBD formulation, ii) the Ancestral RBD based vaccine and iii) bivalent mRNA vaccine BNT162b2 (Pfizer-BioNTech) as control. Results indicated that the Gamma RBD vaccine outperformed the Ancestral RBD formulation in terms of protection

against Omicron BA.5 variant. Interestingly, Gamma RBD derived vaccine induced similar levels of protection against the BA.5 variant as the bivalent mRNA vaccine BNT162b2.

In the revised version of the manuscript, we replaced the graphs of figure 5 with data from the new challenge experiment performed with Omicron BA.5 SARS-CoV-2 variant and the results of the challenge experiment with ancestral Wuhan SARS-CoV-2 has been moved to supplementary section Figure 7.

To summarize, Gamma RBD vaccine-induced protection was observed against 2 different virus variants: Omicron BA.5 and Wuhan SARS-CoV-2. Interestingly, RBD Gamma derived variant induced a similar level of protection against the Omicron BA.5 variant as the mRNA BNT162b2 bivalent vaccine.

Results section was modified accordingly (lines 250-291):

“To determine the vaccine’s ability to protect against SARS-CoV-2 infection, a K-18-hACE2 transgenic mouse model was used (Jiang, Liu et al. 2020, Winkler, Bailey et al. 2020). Transgenic mice of both sexes were immunized on days 0 and 14 with Ancestral RBD plus alum, Gamma RBD plus alum, BNT162b2 bivalent vaccine or placebo as control. Two weeks later, mice were intranasally inoculated with 2×10^4 PFU of Omicron BA.5 SARS-CoV-2. High levels of viral RNA were detected in lung homogenates from placebo group at 3 days post infection (dpi) while in all vaccinated animals the viral genome copies were significantly reduced (Fig. 5a). However, at 5 dpi high viral gene copies were observed at the placebo but only the Gamma RBD plus Alum and bivalent mRNA vaccinated groups had reduced viral RNA at lung tissue (Fig. 5b).

Histopathological analyses of lungs from infected mice of placebo group showed moderate focal to multifocal interstitial pneumonia, peribronchial and perivascular infiltration with areas of moderate type II pneumocyte hyperplasia, hemorrhage, and diffuse edema (Fig. 5b-c). Bronchioles were surrounded by inflammatory cells, mainly neutrophils and mononuclear cells. Also, thickened alveolar walls and reduced alveolar spaces, with loss of epithelial cells and patchy alveolar edema, was observed at 5 dpi. The cumulative pathology score which included the sum of all the pathological findings scores demonstrated moderate to severe histological changes in the placebo group. Histological analysis of lungs obtained from vaccinated mice showed a significant reduction in lung pathology score at 5 dpi (Fig. 5b). Vaccinated mice exhibited only a limited interstitial pneumonia with weak inflammatory infiltrate and hemorrhage, with significantly lower severity in animals immunized with Gamma RBD vaccine or the bivalent BNT162b2 vaccine (Fig. 5c).

In another independent experiment, placebo and Gamma adapted vaccinated K-18-hACE2 mice were challenged intranasally with 2×10^5 PFU of Wuhan SARS-CoV-2 and monitored daily for body weight changes and survival. Immunized mice did not show significant weight loss during the experiment and 66% of the immunized mice survived at 9 dpi whereas all mice in placebo group succumb to infection (Supplementary Fig. 7a-b). High infectious SARS-CoV-2 titers were observed in the lungs and brains of the placebo immunized mice at 5 dpi while low virus titers were detected in lungs and brains of animals vaccinated with Gamma RBD + Alum (Supplementary Fig. 7c). Histopathological changes were analyzed in lung tissue sections and scored according to the severity of the changes. In the placebo group, more significant changes were found in the fibrillar eosinophilic material in the alveolar spaces (edema) and cellular infiltrates in the alveolar septa. Mild or no changes were found in the plump epithelial

cells lining the alveolar septa (type 2 pneumocyte hyperplasia) and infiltration of macrophages in the alveolar spaces. Vaccinated mice presented signs of alveolar cellularity but reduced edema in the lungs. No infiltration or pneumocytes hyperplasia was found in this group (Supplementary Fig. 7d-e).

Thus, immunization with the Gamma RBD-based vaccine induced protection against Ancestral and Omicron BA.5 variant in a mouse model of severe SARS-CoV-2 disease.”

The manuscript will also benefit from some language service. Examples include but not limited to:

Line 43: “One strategy to prevent breakthrough infections is developing safe and effective broad-spectrum vaccines”. Not sure preventing breakthrough infections should be the goal of developing a vaccine as there is no evidence that vaccines can prevent breakthrough infections.

Thank you very much for the suggestion, we reviewed the manuscript language with a native English speaker. In the revised version of manuscript this paragraph was modified according to the reviewer’s suggestion.

“In the context of continuous emergence of SARS-CoV-2 variants of concern (VOCs), one strategy to prevent the severe outcomes of COVID-19 is developing safe and effective broad-spectrum vaccines.” (Lines 46-48. Abstract)

Line 49-51: please fix the grammar of “Vaccine-induced immunity protected K18-hACE2 mice from intranasal challenge with SARS-CoV-2 increasing survival, reducing body weight loss and viral burden in the lungs and brain”.

This sentence was modified accordingly to the new data presented in the revised version of the manuscript. We now show protection against Ancestral and Omicron BA.5 variants and changed the sentence to include both results.

“The Gamma adapted vaccine conferred protection from Ancestral and Omicron BA.5 SARS-CoV-2 variants challenge in mice.” (Lines 56-57. Abstract).

Line 52-55. Please consider rewriting this sentence to smooth out the language. The sentence was modified in the revised manuscript:

“Moreover, Gamma vaccine induced immunogenicity was confirmed after a heterologous booster of different SARS-CoV-2 vaccine platforms. (Lines 58-59. Abstract).

Line 77: Wanning should be waning.

Done.

Thanks again for your comments and suggestions, they improved our work.

At the end of this file, we summarized all new data and analysis included in the revised manuscript to answer reviewer’s comments.

Reviewer #2 (Remarks to the Author):

The paper by Coria et al. investigates a Gamma-adapted SARS-CoV-2 subunit vaccine. RBD from the Gamma variant was formulated in alum and compared with RBD from the

Ancestral Wuhan-Hu-1 strain. The Gamma-adapted vaccine was immunogenic and elicited B and T cell responses in BALB/c mice and partially protected against SARS-CoV-2 challenge in the K18-ACE2 mouse model. It is not clear whether protection was against a Gamma-adapted strain. The vaccine also demonstrated a potent effect as heterologous booster of different vaccine platforms in mice. Surprisingly, compared to the Ancestral RBD, the Gamma-adapted RBD was more immunogenic (in BALB/c mice).

General critique:

It is highly surprising that the Gamma-adapted RBD vaccine elicited higher IgG responses than Ancestral RBD, both against Gamma RBD and Ancestral (Wuhan-Hu-1) RBD. Whilst this may be interesting, the authors do not follow up on these findings. Throughout the manuscript, details on study design, number of repeats and representative stainings are lacking.

Thank you for your comment. After we first analyzed the results that indicated that Gamma RBD is more immunogenic than Ancestral RBD, we were also surprised. However, we repeated these experiments several times obtaining similar results, that support these findings. Moreover, we found that a similar phenomenon has been described for other vaccines containing variant-adapted antigens, like the beta variant (Launay, Cachanado et al. 2022).

As suggested by this reviewer and to further explore this phenomenon, we performed additional experiments comparing the immunogenicity of the gamma and ancestral RBD based vaccines and added these data in the revised version of the manuscript. In addition, and as suggested by the reviewer, we have detailed the number of repeats in each figure legend and representative staining from all flow cytometry experiments are now included in the supplementary information.

Specific comments:

- In figure 2C, the Gamma-adapted RBD vaccine elicited higher IgG responses than Ancestral RBD, both against Gamma RBD and Ancestral RBD. Why was the Gamma-adapted RBD more immunogenic than Ancestral RBD? Could this be due to impurities in the antigen preparations? Did the authors test for pyrogens? The experiment was apparently only performed once. A surprising finding like this requires repeat studies

First, it is important to note that RBD vaccine formulations were produced in a pharmaceutical laboratory (Laboratorio Pablo Cassará) according to good manufacturing practices (GMP). All formulations met the stringent quality requirements for biological products that includes testing of purity, endotoxins, host cell proteins and residual host DNA. We included this information in the results section (lines 115-117):

“Endotoxins, host cell proteins and residual host DNA were examined and complimented the quality requirements for GMP biological products (data not shown).”

Antigens used in the vaccine formulations used in this work showed high levels of purity as is shown in the SDS-PAGE in Fig 1a. Vaccine formulations are routinely tested for endotoxins using the chromogenic limulus amoebocyte lysate assay. Endotoxin levels in the RBD formulations used in this publication are in the table below. We have also added these details on endotoxin determination in the methods section.

“Endotoxin levels of vaccine formulations were determined by *Limulus* amoebocyte assay using Endosafe® Kinetic Chromogenic LAL Reagents (Charles River, Massachusetts, USA). All vaccine formulations contains < 4 EU/ml. Lines 507-510. Methods section.

Lot	Antigen	Dose volume in mice	Endotoxin units (EU/0.1ml)
PrD/DBI/RBDW005-F	Ancestral	0.1	< 0.2
L01	Gamma	0.1	< 0.4
L02	Gamma	0.1	< 0.2

Regarding vaccine elicited immunity, in Fig. 2c we showed that the Gamma-adapted RBD vaccine elicited higher IgG responses than Ancestral RBD, both against Gamma and Ancestral RBD antigen at day 14 post prime immunization (14dpp) but after the second dose these differences were detected against the Gamma RBD (42dpp), as expected.

This experiment was repeated independently in three occasions, using the same dose and frequency of immunization. Serum samples were obtained at day 35 post prime immunization. Results of two other independent experiments are shown below:

As suggested by reviewer 3, in the revised version of the manuscript, we added in Fig. 2c the results of anti- Omicron RBD IgG titers. We observed that the Gamma-adapted RBD vaccine elicited higher IgG responses than Ancestral RBD against the Omicron antigen after the second dose.

To gain some insight into the mechanisms underlying the induction of a superior antibody response by the Gamma-based vaccine compared to the Ancestral version, we further analyzed B and T cell immune responses. As suggested by reviewer 3 we included the analysis of specific B cells, plasmablasts, germinal center B cells and T follicular helper cells in Figure 2e-g. Also, analysis of B cell epitopes in both antigens was performed and is shown in Supplementary Table 2. Results from these new studies were described in the results section.

“To evaluate the mechanisms underlying the superior immunogenicity of the Gamma RBD based vaccine, specific-B cells and plasmablasts were evaluated at day 28 post second dose in splenocytes from immunized animals. Gamma RBD + Alum formulation increased the percentage of Ancestral or Gamma RBD-specific total B cells ($B220^+ CD19^+ IgD^- RBD^+$), IgG^+ B cells ($B220^+ CD19^+ IgD^- IgG^+ RBD^+$) and plasmablasts ($B220^+ CD19^+ IgD^- CD138^+ RBD^+$) while vaccine containing the Ancestral version of RBD

induced only Ancestral RBD-specific B cells and plasmablasts (Fig. 2e and Supplementary Fig. 2).

Germinal center (GC) B cells were also evaluated in spleens of immunized mice. Gamma RBD based-vaccine increased the frequency of GC cells (B220⁺ CD19⁺ IgD⁻ CD95⁺ GL7⁺ cells) compared to placebo group while Ancestral RBD formulation did not (Fig. 2f). No differences in the frequency of T follicular helper cells (Tfh) were found between immunized and placebo groups (Fig. 2g and Supplementary Fig. 2). These results suggest that the superior antibody response of the Gamma adapted vaccine in comparison with Ancestral can be supported by a higher frequency of specific-B cells located at the germinal centers that are involved in the generation of high avidity antibodies.” Lines 143-159.

“To explore the mechanisms that could be involved in the superior antibody response of the Gamma antigen, the primary sequence of RBD of Spike protein from Ancestral or Gamma SARS-CoV-2 were scanned by the IEDB server to predict linear B cell epitopes. Among all the predicted epitopes, those exposed on the surface of S protein with a length between 5-25 residues were selected (Supplementary Table 2). Vaxijen 2.0 was used to calculate antigenicity scores. Among the 5 selected epitopes, the peptide sequence starting at position 496 showed the highest score in both variants (antigenic scores were 0.7632 and 0.8457 for Ancestral and Gamma antigen respectively, Supplementary Table 2). Interestingly, this epitope in the Gamma RBD sequence contains threonine for asparagine substitution in residue 501 (N501Y). In addition, Disco tope analysis of discontinuous epitopes revealed that residues 500 to 506 can be considered as a potential discontinuous epitope in both antigens. Thus, this residue substitution could contribute to the superior immunogenicity of the Gamma RBD formulation.” Lines 193-206.

Discussion of these findings were included in lines 370-382:

“In this work, we first hypothesized and then examined possible mechanisms underlying the superior immunogenicity of the Gamma adapted vaccine. Interestingly, only the Gamma adapted vaccine increased the proportion of germinal center B cells involved in the maturation of high avidity and isotype-switched antibodies required for long-lasting protective immunity. This could be related to the notion that residue substitutions in the Gamma antigen simulate a structure that enhances the spectrum of neutralizing activity. Immunoinformatic tools used for prediction of linear B-cell epitopes in the Ancestral and Gamma RBD sequences revealed that both antigens shared one antigenic epitope that include the N501Y substitution. The presence of the threonine (Y) in the position 501 in the Gamma RBD sequence increased the antigenicity score of this epitope and may explain the superior immunogenicity observed with this formulation. The immunogenicity of this epitope has been previously reported and experimentally confirmed using convalescent sera from COVID-19 patients (Polyjam, Phoolcharoen et al. 2021).”

- Similarly, in figure 2E, the Gamma-adapted RBD vaccine elicited higher CD4⁺ T cell responses (in BALB/c mice) than Ancestral RBD. What could be the reason? Do mutations in the Gamma-adapted RBD give rise to novel immunodominant CD4 T cell epitopes? In this case, it is highly possible that the finding is species-specific or even mouse strain-specific. This should at least be discussed.

Regarding Figure 2E, we have replaced those graphs with data obtained from new experiments. New data regarding T cell responses are now included in Figure 4. To

clarify, in the original Fig 2E, the assay was performed with pooled cells from each group of mice stimulated with mixed RBD peptide pools from Ancestral and Gamma protein. Now we performed new studies with splenocytes from each individual immunized mouse stimulated separately with Ancestral or Gamma RBD peptides. Representative flow cytometry plots from Figure 4a were added as Supplementary Figure 4.

We also analyzed the T cell immune response elicited after immunization in C57BL/6 mice (Supplementary Figure 5). Results from these experiments are described in lines 208-220 of the revised manuscript.

“Cellular immune responses were analyzed in splenocytes from immunized animals by intracellular flow cytometry. The Gamma RBD-based vaccine induced a significant increase in the frequencies of Gamma RBD-specific IFN- γ -, TNF- α - and IL-2-producing CD4⁺ T cells. Moreover, the Gamma RBD adjuvanted vaccine significantly increased the production of IL-2 by CD8⁺ T cells after in vitro stimulation of splenocytes (Fig. 4a, lower panel). The Ancestral version of the vaccine failed to induce cytokine producing CD4⁺ or CD8⁺ T cells after stimulation (Fig. 4a, upper panel). Representative dot plot graphs are shown in Supplementary Fig. 4. Also, RBD-specific CD4⁺ and CD8⁺ T cell immune responses were detected in splenocytes from C57BL/6 mice immunized with the Gamma version of the vaccine (Supplementary Fig. 5). These results demonstrated that the Gamma RBD vaccine induces a higher Ag-specific T cellular immune response than the Ancestral RBD based vaccine formulation in different mouse strains.”

It is well known that genetic background of mouse strains can influence the quantity and quality of the immune responses. Nevertheless, results in two different mouse strains (BALB/c and C57BL/6) revealed that the Gamma-RBD based vaccine was superior to the ancestral RBD based vaccine at inducing T cell responses although the profile of these responses was not identical.

To address the reviewer’s concern if mutations in the Gamma-adapted RBD could give rise to novel immunodominant CD4 T cell epitopes, we performed an in-silico analysis of immunodominant T cell restricted epitopes present in the Gamma and Ancestral version of RBD. We analyzed the MHCI and MHCII restricted epitopes in two haplotypes from the most used laboratory mouse strains (BALB/c and C57BL/6 mice) that were used in this work.

Interestingly, new CD4⁺ and CD8⁺ T cell epitopes in the Gamma adapted RBD were found that may contribute to the increase in T cell responses elicited by this antigen. Results are shown in Supplementary Table 3 and 4 and described in lines 235-247 of the revised version of the manuscript.

“Analysis of immunodominant T cell restricted epitopes present in the Gamma and Ancestral RBD sequences was performed using netMHCpan 4.1 and netMHCpan 3.2. MHCI and MHCII restricted epitopes of 9 residues in length from the two more common mouse haplotypes (d and b) were selected and scored by Vaxijen 2.0. Results indicated that substitution N501Y present in the Gamma RBD sequences generated 3 antigenic epitopes restricted to the I-Ab MHCII haplotype (Supplementary Table 3). These epitopes are not present in the Ancestral RBD sequence. No new epitopes or changes in the score values were found in the I-A^d or I-E^d MHCII haplotypes (BALB/c strain).

Identification of CTL epitopes revealed that new antigenic epitopes carrying the E484K and N501Y were generated in the Gamma sequence in both mouse haplotypes (H-2K^{b/d} and H-2D^{b/d}, Supplementary Table 4). Experimentally, vaccinated mice with the Gamma

RBD adjuvanted vaccine elicited RBD-specific CD4⁺ T cell responses in both mouse strains while RBD-specific CD8⁺ T cells were elicited in C57BL/6 mice.”

- Figure 5: What virus strain was used for challenge? Is it a Gamma virus?

In the challenge experiment of Figure 5 (original manuscript) we used the Ancestral/Wuhan virus strain. This was specified in the methods section: “*mice were challenged intranasally (i.n.) with 10⁵ PFU of SARS-CoV-2 strain WA1/2020*” (Line 639).

In the revised version of the manuscript, we replaced the original figure 5 with data from a new challenge experiment performed with the Omicron BA.5 SARS-CoV-2 variant. In the revised version of the manuscript, results obtained after challenge with the Ancestral Wuhan SARS-CoV-2 in animals immunized with Gamma RBD+Alum are presented in Supplementary Figure 7.

- Figure 5: Whilst lung virus titers appear to be low in vaccinated animals, survival was not 100%. Did remaining mice have signs of lung pathology?

The challenge experiment performed with the Wuhan virus strain was completed at Virginia Tech, and the virus dose inoculated into the model set at 10⁵ PFU. Assessing infectious viral titers by plaque assays, the virus load was undetected in lungs homogenates of vaccinated mice in comparison to the high titers observed with the placebo group indicating vaccine induced protection. However, as the reviewer commented, we do not achieve 100% survival, suggesting there may be pathology lesions in the immunized mice. First, it is important to state that survival differences between groups is statistically significant and, as suggested by the reviewer, histopathology data from lungs at day 5 post infection with the Ancestral SARS-CoV-2 virus are now included in the Supplementary Figure 7. Pathology data revealed that vaccinated mice have some changes in the lung tissue associated with virus infection. Results are detailed in the revised version of the manuscript in lines 273-288:

“In another independent experiment, placebo and Gamma adapted vaccinated K-18-hACE2 mice were challenged intranasally with 2x10⁵ PFU of Wuhan SARS-CoV-2 and monitored daily for body weight changes and survival. Immunized mice did not show significant weight loss during the experiment and 66% of the immunized mice survived at 9 dpi whereas all mice in placebo group succumb to infection (Supplementary Fig. 7a-b). High infectious SARS-CoV-2 titers were observed in the lungs and brains of the placebo immunized mice at 5 dpi while low virus titers were detected in lungs and brains of animals vaccinated with Gamma RBD + Alum (Supplementary Fig. 7c). Histopathological changes were analyzed in lung tissue sections and scored according to the severity of the changes. In the placebo group, more significant changes were found in the fibrillar eosinophilic material in the alveolar spaces (edema) and cellular infiltrates in the alveolar septa. Mild or no changes were found in the plump epithelial cells lining the alveolar septa (type 2 pneumocyte hyperplasia) and infiltration of macrophages in the alveolar spaces. Vaccinated mice presented signs of alveolar cellularity but reduced edema in the lungs. No infiltration or pneumocytes hyperplasia was found in this group (Supplementary Fig. 7d-e).”

- Figure 6: Experimental details are lacking for immunization studies with the different platforms.

Regarding reviewer's question about Figure 6, we added a sentence in methods detailing the route and dose of immunization. Comparative vaccines from this experiment were provided by the Ministry of Health of the Buenos Aires Province, Argentina.

"In booster experiments, eight-week-old female BALB/c mice were i.m inoculated on days 0 and 21 with i) Gamma RBD (10µg/dose) + Alum (100µg/dose) N=5, ii) Comirnaty (1µg/dose) (Pfizer-BioNTech) N=10, iii) BBIBP-CorV (1.3U/dose containing 45µg of Alum) (Sinopharm) N=10, iv) and ChAdOx1 nCoV-19 (3x10⁹ particles/dose) (Oxford-Aztrazeneca) N=10. At day 99, 5 animals per group were boosted with the Gamma RBD (10µg) + Alum (100µg) while the other 5 animals received the homologous vaccine. Blood samples were collected weekly to measure total and neutralizing antibody titers." (Lines 536-542. Methods section).

- Figure 6B-C: Were responses measured against Gamma-specific RBD?

Antibody responses in Fig 6b-c were measured against Ancestral RBD. In the revised version of the manuscript, we detailed this in the figure legend and results section (Lines 318-319).

"Analysis of the neutralizing activity of sera from mice before (pre) and after booster (post) immunization against Ancestral SARS-CoV-2..."

- What was the antigen dose and route of immunization? - What was the dose of alum used for immunization?

The dose of the Gamma RBD vaccine was 10µg of RBD and 100µg of Alum per dose. This corresponds to 1/5 of the human dose tested in the reported phase 1 study (Pasquevich, Coria et al. 2023). All immunizations were delivered intramuscularly. This information is included in Methods section:

"For immunogenicity studies, animals were intramuscularly (i.m) inoculated at day 0 and 14 with i) Gamma RBD (10µg) + Alum (100µg) (N=5), ii) Ancestral RBD (10µg) + Alum (100µg) (N=5) or iii) placebo (Alum alone, N=4)." Lines 528-530. Methods section.

- The supplier of Alum is not stated

Alum in the Gamma RBD vaccine was supplied by CRODA (Croda International plc, United Kingdom) as it is stated in methods section (Lines 505-506).

Thank you for your comments. These suggestions really helped to improve our work.

At the end of this file, we summarized all new data and analysis included in the revised manuscript to answer reviewer's comments.

Reviewer #3 (Remarks to the Author):

This manuscript by Coria et al. reports on an alum-adjuvanted Gamma-adapted recombinant subunit vaccine as a broad-spectrum vaccine candidate. The authors compared antibody responses in mice stimulated with alum-adjuvanted Ancestral and Gamma RBD antigens, showing the Gamma-adapted recombinant subunit vaccine has a potential to elicit neutralizing antibodies against different SARS-CoV-2 variants. The

authors also studied the cellular immune responses stimulated by Gamma-adapted recombinant subunit vaccine. Moreover, the authors demonstrated vaccine-induced protection of K18-hACE2 mice from intranasal viral challenge. This Gamma-adapted recombinant subunit vaccine is also shown to enhance antibody responses as a heterologous booster for different vaccine platforms. Broadly neutralizing antibodies elicited by this Gamma-adapted recombinant subunit vaccine are highly clinically relevant. However, the mechanism of action for this enhanced antibody breadth is not defined. It would add significantly to the paper if the authors were able to obtain any further insights into how this vaccine achieves this effect. In addition, the manuscript has several notable weaknesses that need to be further addressed.

1. In Fig 2C, the authors showed that alum-adjuvanted Gamma RBD improved anti-Gamma IgG and anti-Ancestral IgG more than alum-adjuvanted Ancestral RBD. However, it is unclear that whether alum-adjuvanted Gamma RBD outperforms alum-adjuvanted Ancestral RBD against other SARS-CoV-2 variants. In Fig 3B, the authors showed neutralizing antibodies elicited by alum-adjuvanted Gamma RBD. To make an apple-to-apple comparison, the authors should also include the alum-adjuvanted Ancestral RBD in this figure.

Thank you for this comment. In Fig. 2c we showed that Gamma-adapted RBD vaccine elicited higher IgG responses than Ancestral RBD, both against Gamma RBD and Ancestral RBD 14 days post prime immunization (14dpp) but after the second dose these differences were seen only against the Gamma RBD as expected. In the revised version of the manuscript, we included in Fig. 2c IgG titers against Omicron BA.4/5 RBD. We also observed that Gamma-adapted RBD vaccine elicited higher IgG responses than Ancestral RBD based vaccine against BA.4/5 RBD after the second dose.

In the revised version of the manuscript new experiments and analysis were included to explore the possible mechanisms that may explain a higher immunogenicity of the Gamma RBD antigen compared to Ancestral antigen:

- We performed an in-silico analysis of B cell epitopes in both antigens (Supplementary Table 1, Lines 193-206.). To summarize, we found that the presence of threonine (Y) in position 501 of the Gamma RBD sequence increased the antigenicity score of one restricted B cell epitope and may explain the superior immunogenicity observed with this formulation.
- We analyzed the frequency of specific-B cells, plasmablasts, germinal center B cells and T follicular helper cells in both vaccine groups (Fig. 2e-g, Lines 143-159.). We found that immunization with the Gamma RBD vaccine induced a significant increase in the proportion of germinal center B cells that are involved in the generation of high avidity antibodies. Also, while vaccination with the Gamma RBD based vaccine significantly increased the frequency of Gamma RBD specific- and Ancestral RBD specific B cells, only Ancestral RBD specific B cells were induced upon vaccination with the Ancestral RBD based formulation.

Regarding neutralizing antibody responses showed in Fig. 3b in the original manuscript, we added a new version of this figure comparing Ancestral and Gamma-based vaccines against different VOCs at 42dpp. To visualize and quantify how the different variants relate to each other antigenically, we used the titrations shown in Fig. 3b to construct antigenic maps, where antigens and sera are positioned relative to each other such that the distance between them corresponds to the fold-drop compared to the maximum serum titer. Antigenic distances between Ancestral or Gamma variant and the other

VOCs were reduced in the Gamma RBD vaccinated animals in comparison with the Ancestral vaccine version. Results from this analysis were described in lines 171-179.

“Antigenic cartography was used to explore the different neutralization profiles elicited by the RBD antigens employed. Antigenic maps were made separately for each vaccine using the nAbs titers at day 28 after the second immunization. For the Gamma RBD-adapted formulation, data were more tightly clustered around the Ancestral and Gamma variant (Fig. 3b left panel) in comparison to the Ancestral RBD vaccine (Fig. 3b, right panel). Antigenic distances between Delta, Omicron BA.1 and Omicron BA.5 VOC and the vaccines homologous variants (Ancestral or Gamma) were lower for the vaccine containing the Gamma RBD antigen than the Ancestral RBD formulation (Supplementary Table 1).”

2. Germinal center responses and follicular T responses are often linked to antibody responses. To gain a better understanding of the underlying mechanisms for vaccine-induced cross-reactive antibody responses, the author should investigate these humoral immune responses in both draining lymph nodes and spleens after immunization.

We agree that a detailed analysis of these populations could help to explain the cross-reactive antibody responses induced by the Gamma-adapted vaccine. We therefore conducted new experiments and analyzed germinal center B cells and follicular T cell populations in spleen and draining lymph nodes of immunized mice one month after the second vaccine dose in both vaccine groups (Gamma-RBD and Ancestral-RBD) and the placebo group. Results from these studies using splenocytes are shown in Fig. 2e-g of the revised version of the manuscript and described in lines 143-159. Briefly, we found an increase in the proportion of total and Ag-specific B cells and plasmablasts after immunization of mice with the Gamma version of the vaccine. Interestingly, the Gamma vaccine increased B cells and plasmablasts specific for both the Ancestral and Gamma RBD antigen while the Ancestral based vaccine only was able to increase Ancestral RBD specific-B cells. Moreover, the Gamma RBD adjuvanted vaccine exhibits a higher proportion of germinal center B cells compared to placebo group while the Ancestral vaccine did not. We did not observe any differences in these populations in lymph nodes derived cells.

Gating strategies of flow cytometry analysis are presented in Supplementary Fig. 2 and the methodology described in methods section of the revised manuscript.

The following sentences were added to the revised version of the manuscript:

““To evaluate the mechanisms underlying the superior immunogenicity of the Gamma RBD based vaccine, specific-B cells and plasmablasts were evaluated at day 28 post second dose in splenocytes from immunized animals. Gamma RBD + Alum formulation increased the percentage of Ancestral or Gamma RBD-specific total B cells (B220⁺ CD19⁺ IgD⁻ RBD⁺), IgG⁺ B cells (B220⁺ CD19⁺ IgD⁻ IgG⁺ RBD⁺) and plasmablasts (B220⁺ CD19⁺ IgD⁻ CD138⁺ RBD⁺) while vaccine containing the Ancestral version of RBD induced only Ancestral RBD-specific B cells and plasmablasts (Fig. 2e and Supplementary Fig. 2).

Germinal center (GC) B cells were also evaluated in spleens of immunized mice. Gamma RBD based-vaccine increased the frequency of GC cells (B220⁺ CD19⁺ IgD⁻ CD95⁺ GL7⁺ cells) compared to placebo group while Ancestral RBD formulation did not (Fig. 2f). No differences in the frequency of T follicular helper cells (Tfh) were found between immunized and placebo groups (Fig. 2g and Supplementary Fig. 2). These results suggest that the superior antibody response of the Gamma adapted vaccine in

comparison with Ancestral can be supported by a higher frequency of specific-B cells located at the germinal centers that are involved in the generation of high avidity antibodies.” Lines 143-159.

Discussion of these findings were included in lines 370-376:

“In this work, we first hypothesized and then examined possible mechanisms underlying the superior immunogenicity of the Gamma adapted vaccine. Interestingly, only the Gamma adapted vaccine increased the proportion of germinal center B cells involved in the maturation of high avidity and isotype-switched antibodies required for long-lasting protective immunity. This could be related to the notion that residue substitutions in the Gamma antigen simulate a structure that enhances the spectrum of neutralizing activity.”

3. Fig 2E shows the cellular immune responses elicited by alum-adjuvanted Gamma RBD and alum-adjuvanted Ancestral RBD. It's unclear if there's any statistical significance between the two vaccines in promoting T-cell responses. Specifically, there doesn't seem to be any difference in CD8 T cell responses between the two vaccines.

Regarding T cell immune responses induced by vaccine formulations, graphs from Fig 2e of the original manuscript were replaced by Fig 4a in the revised version of the manuscript. In the original Fig 2e, results were performed with pooled cells from each group of mice stimulated with mixed RBD peptides pools from Ancestral and Gamma protein. We performed new experiments with splenocytes from each individual immunized mouse stimulated separately with either Ancestral or Gamma RBD peptides to obtain more accurate and variant specific conclusions. Briefly, the Gamma RBD based vaccine elicited cytokine producing CD4⁺ and CD8⁺ T cells specific for the Gamma antigen while the Ancestral vaccine failed to induce T cell immune responses. Similar results were obtained in the C57BL/6 mouse strain.

Results from these experiments are described in lines 208-220:

“Cellular immune responses were analyzed in splenocytes from immunized animals by intracellular flow cytometry. The Gamma RBD-based vaccine induced a significant increase in the frequencies of Gamma RBD-specific IFN- γ -, TNF- α - and IL-2-producing CD4⁺ T cells. Moreover, the Gamma RBD adjuvanted vaccine significantly increased the production of IL-2 by CD8⁺ T cells after in vitro stimulation of splenocytes (Fig. 4a, lower panel). The Ancestral version of the vaccine failed to induce cytokine producing CD4⁺ or CD8⁺ T cells after stimulation (Fig. 4a, upper panel). Representative dot plot graphs are shown in Supplementary Fig. 4. Also, RBD-specific CD4⁺ and CD8⁺ T cell immune responses were detected in splenocytes from C57BL/6 mice immunized with the Gamma version of the vaccine (Supplementary Fig. 5). These results demonstrated that the Gamma RBD vaccine induces a higher Ag-specific T cellular immune response than the Ancestral RBD based vaccine formulation in different mouse strains.”

It is well known that genetic background of mouse strains can influence the quantity and quality of the immune responses. Nevertheless, results in two different mouse strains (BALB/c and C57BL/6) revealed that the Gamma-RBD based vaccine was superior to the ancestral RBD based vaccine in the induction of T cell responses although the profile of these responses are not identical.

In response to reviewer 2 comment, we also performed an in-silico analysis of immunodominant T cell restricted epitopes present in the Gamma and Ancestral version

of RBD. We analyzed the MHC I and MHC II restricted epitopes in two haplotypes from the most used laboratory mouse strains (BALB/c and C57BL/6).

Interestingly, new CD4⁺ and CD8⁺ T cell epitopes in the Gamma adapted RBD were found that may contribute to the increase in T cell responses elicited by this antigen. Results are shown in Supplementary Table 3 and 4 and described in lines 2365-247:

“Analysis of immunodominant T cell restricted epitopes present in the Gamma and Ancestral RBD sequences was performed using netMHCpan 4.1 and netMHCpan 3.2. MHC I and MHC II restricted epitopes of 9 residues in length from the two more common mouse haplotypes (d and b) were selected and scored by Vaxijen 2.0. Results indicated that substitution N501Y present in the Gamma RBD sequences generated 3 antigenic epitopes restricted to the I-A^b MHC II haplotype (Supplementary Table 3). These epitopes are not present in the Ancestral RBD sequence. No new epitopes or changes in the score values were found in the I-A^d or I-E^d MHC II haplotypes (BALB/c strain).

Identification of CTL epitopes revealed that new antigenic epitopes carrying the E484K and N501Y were generated in the Gamma sequence in both mouse haplotypes (H-2K^{b/d} and H-2D^{b/d}, Supplementary Table 4). Experimentally, vaccinated mice with the Gamma RBD adjuvanted vaccine elicited RBD-specific CD4⁺ T cell responses in both mouse strains while RBD-specific CD8⁺ T cells were elicited in C57BL/6 mice.”

4. In Fig 3A, the authors showed long-lasting antibody responses elicited by alum-adjuvanted Gamma RBD. Does it correlate with plasma cells accumulating in the bone marrow?

It would be ideal to measure plasma cells at 253dpp in the bone marrow, unfortunately we do not have samples from that experiment and to repeat it will take a considerably long time. However and following the reviewer’s suggestions, we performed the analysis of long-lived plasma cells (LLPCs) in bone marrow cells from immunized mice at day 42 post prime immunization. We found that total and IgG⁺ RBD-specific LLPCs populations were enhanced in the vaccinated animals compared to placebo group.

Results are described in lines 187-192:

“Analysis of long-lived plasma cells (LLPCs) in the bone marrow of immunized mice at day 42 dpp indicates that the frequencies of total (B220⁺ CD138⁺) and isotype-switched RBD-specific LLPCs (B220⁺ CD138⁺ IgG⁺ RBD⁺) were higher in vaccinated mice with the Gamma subunit vaccine compared to placebo group (Supplementary Fig. 3). These results further explain the induction of a broader and long-lasting nAbs response by the Gamma RBD adjuvanted formulation.

We also added a sentence in the discussion section lines 401-407:

“In mice, the Gamma adapted vaccine induced a long-lasting antibody response with neutralizing activity against different VOCs up to 253 days after the last immunization. High-affinity antibodies that provide durable protective immunity are produced by long-lived bone marrow plasma cells differentiated from germinal center B cells. Indeed, after immunization with the Gamma vaccine formulation total and RBD-specific LLPCs in bone marrow were observed indicating that long-term antibodies could be secreted from those cells.”

5. The authors characterized IL-5 cytokine in Fig 4A. The authors should also

characterize other crucial Th2 cytokines including IL-4, IL-10, IL-13, and IL-21 as they are often highly correlated with vaccine-induced humoral immune responses and antibody responses.

Following the reviewer's suggestion we performed a multiplex bead-based assay that allowed the simultaneous quantification of 12 cytokines in cell culture supernatant samples. Splenocytes from immunized mice were stimulated with culture media, Gamma RBD or Ancestral RBD and then cytokine secretion was evaluated. Interestingly, while the Gamma RBD vaccine induced mixed Th1/Th2 responses with production of IFN- γ , TNF- α , IL-2, IL-6, IL-10, IL-4 and IL-5, the Ancestral RBD vaccine induced the production of predominantly Th2 related cytokines (IL-2, IL-6, IL-13, IL-4 and IL-5). In addition, Th17 and Th9 and Th22 related cytokines were also induced by the Gamma RBD + Alum formulation."

These results are shown in Fig.4b of the revised version of the manuscript and described in results section in lines 227-234:

"Evaluation of Th cytokine profile was performed by multiplex assay in supernatants from splenocytes of immunized mice stimulated with Ancestral or Gamma RBD antigen. Results revealed that the Gamma RBD vaccine induced mixed Ag-specific Th1/Th2 responses with production of IFN- γ , TNF- α , IL-2, IL-6, IL-10, IL-4 and IL-5 (Fig. 4b). On the other hand, predominantly Th2 related cytokines were produced (IL-2, IL-6, IL-13, IL-4 and IL-5, Fig. 4b) after immunization with the Ancestral RBD vaccine. Interestingly, Th17 and Th9 and Th22 related cytokines (IL-17A/F, IL-9 and IL-22) were induced by the Gamma RBD + Alum formulation."

6. The label in the right panel of Fig 4B is not correct. It should be "CD8 T cell responses in spleen".

This figure Fig.4B is no longer in the manuscript.

7. To make an apple-to-apple comparison, the appropriate control in Fig 4 should be alum-adjuvanted Ancestral RBD instead of a placebo.

As suggested by the reviewer, a new experiment was performed to compare the T cell responses from both vaccine formulations (Gamma RBD vs. Ancestral RBD). Results shown in the original manuscript in Fig.4 with the Gamma RBD vaccine were replaced in the revised version with new data derived from both vaccines and is shown in Supplementary Figure 5.

These results were described in lines 221-226 of the results section:

"In addition, analysis of polyfunctional T cells showed that responses elicited by the Gamma RBD based vaccine had a predominance of specific double-positive (IFN- γ ⁺ and TNF- α ⁺) and triple-positive (IFN- γ ⁺ TNF- α ⁺ and IL-2⁺) cytokine producing CD4⁺ T cells (Supplementary Fig. 6). These results demonstrated that vaccine containing Gamma RBD antigen induces a superior Ag-specific Th1 cellular immune response than RBD derived from the Ancestral SARS-CoV-2."

A sentence was included in the discussion section in lines 414-420:

“Here, we have provided a comprehensive assessment of the T cell response to both vaccine versions. Gamma RBD + alum vaccination elicited a mixed Th1/Th2 T cell response while the Ancestral based vaccine induced a predominantly Th2 response. Analysis of polyfunctionality on vaccine elicited T cells suggests that the major proportion of T cells produce two cytokines (IFN- γ and TNF- α) or three cytokines (IFN- γ , TNF- α and IL-2). Polyfunctional Th1 responses were reported after human vaccination with adenoviral ChAdOx1-S and the mRNA BNT162b2 vaccine (Guerrera, Picozza et al. 2021, Swanson, Padilla et al. 2021)”

8. To demonstrate the benefits of broad antibody responses elicited by alum-adjuvanted Gamma RBD vaccine, the author should challenge mice with a different strain of the SARS-CoV-2 virus. Also, the appropriate control in this figure should be alum-adjuvanted Ancestral RBD instead of a placebo.

Following the reviewer’s suggestion, we performed new animal challenge studies using the Omicron BA.5 SARS-CoV-2 variant. Briefly, hACE-2 transgenic mice were immunized at days 0 and 14 with either the Gamma or the Ancestral version of the vaccine. Also, we also included another group of mice that was immunized with the BNT162b2 bivalent vaccine. Vaccinated mice were compared to a placebo group. Two weeks later, animals were intranasally inoculated with the Omicron BA.5 VOC and after 3 or 5 days post infection (dpi) viral loads were determined by qPCR and histopathology in the lung tissues was evaluated. Results showed that all vaccinated animals had reduced viral loads in the lungs at 3 dpi but only the Gamma subunit vaccine and the bivalent mRNA vaccine maintained these low viral loads at 5 dpi. Histopathology analysis also demonstrated that both the Gamma-based vaccine and the bivalent BNT162b2 vaccine are the best at reducing the lesions observed in the placebo infected mice. Results from these experiments were added in the revised manuscript replacing the original Figure 5 (now presented as Supplementary Fig. 7). These findings were described in lines 250-291 of the result section in the revised version of the manuscript.

“To determine the vaccine’s ability to protect against SARS-CoV-2 infection, a K-18-hACE2 transgenic mouse model was used (Jiang, Liu et al. 2020, Winkler, Bailey et al. 2020). Transgenic mice of both sexes were immunized on days 0 and 14 with Ancestral RBD plus alum, Gamma RBD plus alum, BNT162b2 bivalent vaccine or placebo as control. Two weeks later, mice were intranasally inoculated with 2×10^4 PFU of Omicron BA.5 SARS-CoV-2. High levels of viral RNA were detected in lung homogenates from placebo group at 3 days post infection (dpi) while in all vaccinated animals the viral genome copies were significantly reduced (Fig. 5a). However, at 5 dpi high viral gene copies were observed at the placebo but only the Gamma RBD plus Alum and bivalent mRNA vaccinated groups had reduced viral RNA at lung tissue (Fig. 5b).

Histopathological analyses of lungs from infected mice of placebo group showed moderate focal to multifocal interstitial pneumonia, peribronchial and perivascular infiltration with areas of moderate type II pneumocyte hyperplasia, hemorrhage, and diffuse edema (Fig. 5b-c). Bronchioles were surrounded by inflammatory cells, mainly neutrophils and mononuclear cells. Also, thickened alveolar walls and reduced alveolar spaces, with loss of epithelial cells and patchy alveolar edema, was observed at 5 dpi. The cumulative pathology score which included the sum of all the pathological findings scores demonstrated moderate to severe histological changes in the placebo group. Histological analysis of lungs obtained from vaccinated mice showed a significant reduction in lung pathology score at 5 dpi (Fig. 5b). Vaccinated mice exhibited only a

limited interstitial pneumonia with weak inflammatory infiltrate and hemorrhage, with significantly lower severity in animals immunized with Gamma RBD vaccine or the bivalent BNT162b2 vaccine (Fig. 5c).

In another independent experiment, placebo and Gamma adapted vaccinated K-18-hACE2 mice were challenged intranasally with 2×10^5 PFU of Wuhan SARS-CoV-2 and monitored daily for body weight changes and survival. Immunized mice did not show significant weight loss during the experiment and 66% of the immunized mice survived at 9 dpi whereas all mice in placebo group succumb to infection (Supplementary Fig. 7a-b). High infectious SARS-CoV-2 titers were observed in the lungs and brains of the placebo immunized mice at 5 dpi while low virus titers were detected in lungs and brains of animals vaccinated with Gamma RBD + Alum (Supplementary Fig. 7c). Histopathological changes were analyzed in lung tissue sections and scored according to the severity of the changes. In the placebo group, more significant changes were found in the fibrillar eosinophilic material in the alveolar spaces (edema) and cellular infiltrates in the alveolar septa. Mild or no changes were found in the plump epithelial cells lining the alveolar septa (type 2 pneumocyte hyperplasia) and infiltration of macrophages in the alveolar spaces. Vaccinated mice presented signs of alveolar cellularity but reduced edema in the lungs. No infiltration or pneumocytes hyperplasia was found in this group (Supplementary Fig. 7d-e).

Thus, immunization with the Gamma RBD-based vaccine induced protection against Ancestral and Omicron BA.5 variant in a mouse model of severe SARS-CoV-2 disease.”

Thank you for your comments and suggestions that helped to improve this work.

To summarize additional experimental data and analysis included in the manuscript:

- i) Ancestral RBD based vaccine formulation was added as control in new immunogenicity and challenge/protection studies in mice. Inclusion of this group allowed comparison of both vaccine antigens (Gamma and Ancestral).
- ii) As suggested by the reviewers we explore the possible mechanism underlying the superior immunogenicity of the Gamma RBD vaccine formulation in comparison with the ancestral RBD formulation:
 - a. We analyzed the frequency of B cells, plasmablasts, germinal center B cells and T follicular helper cells in both vaccinated groups. We found that immunization with the Gamma RBD vaccine induced a significant increase in the proportion of germinal center B cells that are involved in the generation of high avidity antibodies. Vaccination with the Gamma RBD formulation significantly increased the frequency of Gamma RBD specific- and Ancestral RBD specific B cells, while Ancestral RBD based formulation induced only Ancestral RBD specific B cells.
 - b. We performed an in-silico analysis of B cell epitopes in both antigens. We found that one residue substitution (N501Y) present in the Gamma RBD sequence increased the antigenicity score of one restricted B cell epitope and may explain the superior immunogenicity observed with this formulation.

c. We present a comprehensive assessment of the T cell response induced by both vaccine versions in two mice strains. Gamma RBD + alum vaccination elicited a mixed Th1/Th2 T cell response while the Ancestral based vaccine induced a predominantly Th2 response. Analysis of TNF- α , IL-2 and IFN- γ production on Gamma vaccine elicited T cells indicated that this vaccine formulation induced a main proportion of polyfunctional T cells.

d. We performed an in-silico analysis of immunodominant T cell restricted epitopes present in the Gamma and Ancestral version of RBD. New epitopes were generated in the Gamma RBD sequence that may contribute to the superior specific-T cell responses observed.

iv) We included histopathology analysis of lung tissue of animals infected with the Wuhan SARS-CoV-2 strain showed in the original manuscript. Moreover, new challenge studies using Omicron BA.5 variant was performed in mice vaccinated with Gamma or Ancestral RBD vaccine formulation. We added a group of mice vaccinated with the bivalent mRNA vaccine BNT162b2 as control. Protection studies confirmed that the immunity induced by the Gamma-derived RBD vaccine candidate can protect mice from a nasal challenge of Ancestral and Omicron BA.5 SARS-CoV-2 virus. Indeed, the Gamma vaccine induced protection against Omicron BA.5 variant in a similar extent than bivalent mRNA vaccine BNT162b2 while ancestral RBD did not.

Reviewers' Comments:

Reviewer #1:

Remarks to the Author:

The authors made efforts to address concerns raised from the previous round of review. However, I still have a couple comments:

Figure 5: Please also provide weight loss and survival curves as you did in supplementary figure 7.

The finding that Gamma variant RBD is more immunogenic is somewhat against what has been reported. At the very beginning of the pandemic, both mRNA manufacturers as well as multiple academic groups have tested RBD as immunogen and then gave it up due to poor performance. Don't the authors agree that updated vaccines with newer variant sequences would easily outperform the Gamma RBD vaccine as a booster? What would be the significance of the Gamma RBD vaccine then?

Reviewer #2:

Remarks to the Author:

The authors have addressed most of my comments sufficiently and the additional (potential) mechanisms for why the gamma RBD-based vaccine is more immunogenic than ancestral RBD are appreciated. However, I still have some concerns, particularly to some of the new data that have been added.

Line 153: The statement that the ancestral RBD-based vaccine does not elicit GC responses is not well-supported by data. In Fig 2F, the difference between gamma RBD vaccine and ancestral RBD vaccine is minimal.

Line 213: Data in fig 4 are not very convincing. Specifically, there seems to be an increase in CD4 T cells producing the measured cytokines both after gamma and ancestral RBD immunization. The authors have analyzed immunodominant T cell restricted epitopes present in the gamma and ancestral RBD, suggesting that new MHCII epitopes are generated in gamma RBD. However, in light of this, it is surprising that restimulation with gamma RBD gave higher cytokine responses (particularly IFN-gamma) even after immunization with the ancestral RBD. Also, the background levels (medium control) were high for the ancestral RBD vaccine (and lower for the gamma RBD vaccine), which likely explains why there was a significant difference between the gamma RBD and medium, but not for ancestral RBD versus medium. Increasing the number of mice would be necessary to draw conclusions on this (There were only four mice per group in the study).

Line 232: The statements that gamma RBD elicited a mixed Th1/Th2 profile in contrast to ancestral RBD is not well-supported by data. E.g. it is stated that gamma-adapted RBD induced IFN- γ , TNF- α , IL-2, IL-6, IL-10, IL-4 and IL-5, whilst ancestral RBD vaccine elicited predominantly Th2 related cytokines (IL-2, IL-6, IL-13, IL-4 and IL-5). However, the levels of IL-4, IL-5 and IL-13 were similar between the groups and the differences in IFN-gamma are minimal. In general, for the T cell data, the comparisons should preferably be between the gamma and ancestral vaccine and not a comparison to the medium control.

Supplementary figure 4: Related to my previous comments about figure 4, the representative flow cytometry plots suggest that there were a substantial frequency of dead cells in the ICS data. The authors should add percentages of cells in the different gates. Overall, this reviewer is not convinced of the quality of the ICS data.

Line 608: Please add details on length of RBD peptides used for restimulation. What was the peptide concentration used for restimulation?

Reviewer #3:

Remarks to the Author:

The revised version of the manuscript has been improved significantly.

However, the authors stated that the superior antibody response of the Gamma-adapted vaccine in comparison with Ancestral can be supported by a higher frequency of specific-B cells located at the germinal centers that are involved in the generation of high avidity antibodies. But Fig 2f clearly showed there's no difference in germinal center B cells between the Gamma-adapted vaccine group and the Ancestral group. The authors are recommended to compare antigen-specific (RBD+) germinal center B cells within these two groups in order to support their statement.

Point-by-point response to the reviewers' comments

We would like to thank the reviewers for their valuable comments, which have further improved our manuscript.

Please find below a point-by-point response to the reviewers' comments. Reviewers' comments are written in black and responses in blue.

REVIEWER COMMENTS

Reviewer #1 (Remarks to the Author):

The authors made efforts to address concerns raised from the previous round of review. However, I still have a couple comments:

Figure 5: Please also provide weight loss and survival curves as you did in supplementary figure 7.

Regarding the challenge experiment with Omicron BA.5 variant, it is important to mention that it was reported previously that Omicron and its lineages present an attenuated phenotype in hACE2 transgenic mice and hamsters, so the severity of the disease is reduced compared to previous variants (Halfmann, Iida et al. 2022, Uraki, Halfmann et al. 2022). In most of the cases, infection with the BA.5 isolates does not cause substantial changes in body weight and animals do not succumb to infection. In the revised version of the manuscript, we added as supplementary figure 7 the graph showing the weight loss and survival until day 5 of the animals challenged with Omicron BA.5 variant. None of the animals died after the infection and no significant weight loss was observed in any group.

In the revised manuscript, in the results section, we added a paragraph describing these results:

Lines 265-267: *"In agreement to the reduced pathogenicity reported for this virus variant, infected animals did not lose weight neither succumb to infection (Supplementary Fig. 7)."*

The finding that Gamma variant RBD is more immunogenic is somewhat against what has been reported. At the very beginning of the pandemic, both mRNA manufacturers as well as multiple academic groups have tested RBD as immunogen and then gave it up due to poor performance. Don't the authors agree that updated vaccines with newer variant sequences would easily outperform the Gamma RBD vaccine as a booster? What would be the significance of the Gamma RBD vaccine then?

Our findings revealed that the Gamma RBD variant outperformed the ancestral SARS-CoV-2 RBD formulation in terms of immunogenicity. We do not have knowledge of previous reports using Gamma variant antigens in COVID-19 vaccines.

Regarding the use of RBD in mRNA vaccines, in 2020 Pfizer-BioNTech developed two lipid nanoparticle-formulated, nucleoside-modified RNA vaccine candidates: BNT162b1, which encodes a secreted trimerized SARS-CoV-2 receptor-binding domain -RBD- from ancestral SARS-CoV-2 strain; or BNT162b2, which encodes a membrane-anchored SARS-CoV-2 full-length spike, stabilized in the prefusion conformation. In the preclinical studies, BNT162b1 and BNT162b2 candidates induced favorable viral antigen specific CD4⁺ and CD8⁺T cell responses, high levels of neutralizing antibody in various animal species, and beneficial protective effects in a primate SARS-CoV-2 challenge model. In

clinical studies, BNT162b2 was associated with a lower incidence and severity of systemic reactions than BNT162b1, particularly in older adults. In both younger and older adults, the two vaccine candidates elicited similar dose-dependent SARS-CoV-2–neutralizing geometric mean titers, which were similar to or higher than the geometric mean titer of a panel of SARS-CoV-2 convalescent serum samples (Walsh, Frenck et al. 2020). Based on clinical safety results, the company decided to go further with BNT162b2. They never pointed out that there were differences in immunogenicity between both vaccines. In addition, it has been reported the great potential of RBD as an antigen in COVID-19 vaccines (Kleanthous, Silverman et al. 2021). Currently, 20 subunit COVID-19 vaccines have been approved for use (COVID Vaccine Tracker), of them 12 contain RBD as antigen (Dai, Gao et al. 2022, Eugenia-Toledo-Romani, Verdecia-Sanchez et al. 2022, Corominas, Garriga et al. 2023)

The significant contribution of this work relies in that we provide proof of concept regarding the importance of the Gamma variant antigen in a vaccine formulation to influence the immunogenicity of the vaccine formulation, such as T cell responses and the breath of neutralizing antibodies against different SARS-CoV-2 VOCs.

We agree with the reviewer that the emergence of new virus variants will require recurrent vaccine updates, nevertheless in this work we highlighted the use of the Gamma RBD variant to be used in multivalent or multivariant vaccines to induce a broader response. In fact, we have recently demonstrated in a phase 3 trial that a bivalent RBD vaccine containing Gamma and Omicron BA4/5 variants induced a better neutralizing activity against Omicron BA.5 than the monovalent vaccine itself (unpublished results). This is why the bivalent vaccine obtained the authorization by the Argentina regulatory agency to be used in people over 18 years old as booster.

We added a sentence about use of Gamma in bivalent vaccines and slightly modified the final statement of the discussion session.

Lines 448-453: “Recently, the monovalent Gamma and bivalent (Gamma-Omicron BA.4/5) vaccines have been registered for use as booster in Argentina becoming the first local vaccine against SARS-CoV-2 produced in the region.

This work highlights the value of a Gamma variant-adapted vaccine to induce higher and broader immune responses against SARS-CoV-2 as primary or booster vaccination in monovalent or multivalent updated vaccines.”

Reviewer #2 (Remarks to the Author):

The authors have addressed most of my comments sufficiently and the additional (potential) mechanisms for why the gamma RBD-based vaccine is more immunogenic than ancestral RBD are appreciated. However, I still have some concerns, particularly to some of the new data that have been added.

Line 153: The statement that the ancestral RBD-based vaccine does not elicit GC responses is not well-supported by data. In Fig 2F, the difference between gamma RBD vaccine and ancestral RBD vaccine is minimal.

Thanks for your comment, Figure 2f showed that there is significant difference in the percentage of GC cells between the Gamma vaccine and the placebo group (*p=0.00349). Besides, it is stated in the graph that there is not a significant difference between the gamma and ancestral based vaccines (ns). Results section was modified to clarify the absence of significant differences between both vaccine groups. Conclusions about this result were modified accordingly in results and abstract sections.

Lines 150-153: *“Germinal center (GC) B cells were also evaluated in spleens of immunized mice. Gamma RBD based-vaccine increased the frequency of GC cells (B220⁺ CD19⁺ IgD⁻ CD95⁺ GL7⁺ cells) compared to placebo group while no significant differences were found compared to the Ancestral RBD formulation (Fig. 2f).”*

Line 213: Data in fig 4 are not very convincing. Specifically, there seems to be an increase in CD4 T cells producing the measured cytokines both after gamma and ancestral RBD immunization. The authors have analyzed immunodominant T cell restricted epitopes present in the gamma and ancestral RBD, suggesting that new MHCII epitopes are generated in gamma RBD. However, in light of this, it is surprising that restimulation with gamma RBD gave higher cytokine responses (particularly IFN-gamma) even after immunization with the ancestral RBD. Also, the background levels (medium control) were high for the ancestral RBD vaccine (and lower for the gamma RBD vaccine), which likely explains why there was a significant difference between the gamma RBD and medium, but not for ancestral RBD versus medium. Increasing the number of mice would be necessary to draw conclusions on this (There were only four mice per group in the study).

Thanks for your comment. ICS data showed in the first revised version of the manuscript were obtained using thawed frozen splenocytes, this can explain the high percentage of dead cells and high background levels of the ICS data. To improve the viability of cells we performed new experiments, we immunized mice (n=6/group) and performed the ICS experiment using fresh splenocytes. New results are shown in Fig. 4a replacing previous data from the first revised version of the manuscript. In this new data, the number of mice per group was increased and background signal is more consistent between groups allowing a better interpretation of data. New data is very similar to the previous one regarding CD4⁺ T cells responses the main change was observed in CD8⁺ T cells where the reduction of background signal reveal significant differences in the cytokines production in both vaccinated groups (percentages are lower in comparison to CD4⁺ cytokine producing T cells). Main differences between groups were found in the induction of IFN- γ producing CD4⁺ T cells revealing that the Gamma vaccine outperformed the Ancestral vaccine version. We also made a new analysis of polyfunctional T cells showed in Supplementary Figure 6. New data was very similar to previous one, with a predominance of specific double-positive (IFN- γ ⁺ and TNF- α ⁺) cytokine producing CD4⁺ T cells in the Gamma based vaccine after stimulation with the antigen.

Results section was modified accordingly to new data:

Lines 210-225: *“Cellular immune responses were analyzed in splenocytes from immunized animals by intracellular flow cytometry. The Gamma RBD-based vaccine induced a significant increase in the frequencies of Gamma RBD-specific IFN- γ -, TNF- α - and IL-2-producing CD4⁺ T cells after restimulation with the Gamma or Ancestral antigen (Fig. 4a, upper panel) while the ancestral version of the vaccine failed to induce IFN- γ in CD4⁺ T cells and TNF- α was only observed after Gamma antigen restimulation. Both vaccines have a similar performance at inducing CD8⁺ T cells after stimulation (Fig. 4a, lower panel). Representative dot plot graphs are shown in Supplementary Fig. 4. Also, RBD-specific CD4⁺ and CD8⁺ T cell immune responses were detected in splenocytes from C57BL/6 mice immunized with the Gamma version of the vaccine (Supplementary Fig. 5). These results demonstrated that the Gamma RBD vaccine induces mixed Th1/Th2 Ag-specific cellular immune responses while the Ancestral RBD based vaccine formulation induces cellular immune responses predominantly with a Th2 profile.”*

In addition, analysis of polyfunctional T cells showed that responses elicited by the Gamma RBD based vaccine had a predominance of specific double-positive (IFN- γ ⁺ and

TNF- α) cytokine producing CD4⁺ T cells in comparison with the ancestral vaccine version (Supplementary Fig. 6). These results support that vaccine containing Gamma RBD antigen induces a superior Ag-specific Th1 cellular immune response than RBD derived from the Ancestral SARS-CoV-2.”

Discussion section was also modified accordingly to new results.

Lines 414-419: “Here, we have provided a comprehensive assessment of the T cell response to both vaccine versions. Gamma RBD + alum vaccination elicited a mixed Th1/Th2 cell response while the Ancestral based vaccine induced a predominantly Th2 response. Analysis of polyfunctionality on vaccine elicited T cells indicated that the Gamma vaccine have a major proportion of T cells that produce two cytokines (IFN- γ and TNF- α) upon Ag stimulation.”

Regarding your comment about the prediction of T cells epitopes in both RBD sequences, we found that new antigenic epitopes were generated in the Gamma sequence restricted to both MHC (I and II) while the most important differences between vaccines were observed associated to CD4⁺ T cells. We hypothesized that antigenic epitopes restricted to MHC I shared by both antigens are enough to induce robust CD8⁺ T cells responses and new epitopes restricted to MHC II in the Gamma RBD could be determinant to enhance the quality of CD4⁺ T cell responses.

Line 232: The statements that gamma RBD elicited a mixed Th1/Th2 profile in contrast to ancestral RBD is not well-supported by data. E.g. it is stated that gamma-adapted RBD induced IFN- γ , TNF- α , IL-2, IL-6, IL-10, IL-4 and IL-5, whilst ancestral RBD vaccine elicited predominantly Th2 related cytokines (IL-2, IL-6, IL-13, IL-4 and IL-5). However, the levels of IL-4, IL-5 and IL-13 were similar between the groups and the differences in IFN- γ are minimal. In general, for the T cell data, the comparisons should preferably be between the gamma and ancestral vaccine and not a comparison to the medium control.

In Fig 4b, we performed the statistical analysis comparing to medium control and between vaccine groups. It is indicated with stars when we found a statistical significance between comparisons. Not significant differences were not indicated in the graph to make simply to interpretate. We found statistical differences between vaccine groups in IL-13 and IL-5 as is detailed in the graphs. We agree with the reviewer that levels of Th2 associated cytokines are high in both vaccine groups, the main difference between groups is that the Gamma based vaccine can induce significant levels of IFN- γ and TNF- α while the ancestral based vaccine cannot. These results together with the ICS data allow us to conclude that the Gamma RBD antigen induces a mixed Th1/Th2 cell response while the lack of significant IFN- γ production (assessed by multiplex and ICS) by the Ancestral antigen suggest a predominant Th2 profile.

Supplementary figure 4: Related to my previous comments about figure 4, the representative flow cytometry plots suggest that there were a substantial frequency of dead cells in the ICS data. The authors should add percentages of cells in the different gates. Overall, this reviewer is not convinced of the quality of the ICS data.

To answer the comments about figure 4, new experiments were performed, and new ICS results are shown in the revised version of the manuscript. Previous data was obtained with fresh splenocytes. In this new data the frequency of dead cells was reduced considerably (around 65-85% of live cells). It is important to mention that in general the methodology of ICS reduces the viability of cells, overnight stimulation together with the

use of brefeldin (inhibitor of protein transport from the endoplasmic reticulum to Golgi) can induce certain degree of cell death. For this reason, we always use a live cell marker (Zombie dye) that allow us to exclude dead cells in the analysis. Given that cell death is constant among the groups and that we use a dead cell marker, we can be confident about presented data. New gating strategy (with gate frequencies) and representative dot plots for each group were replaced in Supplementary Figure 4.

Line 608: Please add details on length of RBD peptides used for restimulation. What was the peptide concentration used for restimulation?

In the work, cells were stimulated with 0.2µg/peptide of a pool of 53 peptides derived from a peptide scan (15mers with 11 aa overlap) through Receptor Binding Domain of Spike Glycoprotein (Swiss-Prot ID: P0DTC2, region: 319-541) of SARS-CoV-2 from the Wuhan or Gamma variant (Lineage P.1, B.1.1.28.1)

We added these details in the methods section (Lines 611-616).

Reviewer #3 (Remarks to the Author):

The revised version of the manuscript has been improved significantly.

However, the authors stated that the superior antibody response of the Gamma-adapted vaccine in comparison with Ancestral can be supported by a higher frequency of specific-B cells located at the germinal centers that are involved in the generation of high avidity antibodies. But Fig 2f clearly showed there's no difference in germinal center B cells between the Gamma-adapted vaccine group and the Ancestral group. The authors are recommended to compare antigen-specific (RBD+) germinal center B cells within these two groups in order to support their statement.

Thanks for your comment, Figure 2f showed that there is significant difference in the percentage of GC cells between the Gamma vaccine and the placebo group (*p=0.00349). Besides, it is stated in the graph that there is not a significant difference between the gamma and ancestral based vaccines (ns). Results section was modified to clarify the absence of significant differences between both vaccine groups. Conclusions about this result were modified accordingly in results and abstract sections.

Lines 150-153: *“Germinal center (GC) B cells were also evaluated in spleens of immunized mice. Gamma RBD based-vaccine increased the frequency of GC cells (B220⁺ CD19⁺ IgD⁻ CD95⁺ GL7⁺ cells) compared to placebo group while no significant differences were found compared to the Ancestral RBD formulation (Fig. 2f).”*

Corominas, J., et al. (2023). "Safety and immunogenicity of the protein-based PHH-1V compared to BNT162b2 as a heterologous SARS-CoV-2 booster vaccine in adults vaccinated against COVID-19: a multicentre, randomised, double-blind, non-inferiority phase IIb trial." The Lancet Regional Health - Europe **28**: 100613.

Dai, L., et al. (2022). "Efficacy and Safety of the RBD-Dimer–Based Covid-19 Vaccine ZF2001 in Adults." New England Journal of Medicine **386**(22): 2097-2111.

Eugenia-Toledo-Romani, M., et al. (2022). "Safety and immunogenicity of anti-SARS CoV-2 vaccine SOBERANA 02 in homologous or heterologous scheme: Open label phase I and phase IIa clinical trials." Vaccine **40**(31): 4220-4230.

Halfmann, P. J., et al. (2022). "SARS-CoV-2 Omicron virus causes attenuated disease in mice and hamsters." Nature **603**(7902): 687-692.

Kleanthous, H., et al. (2021). "Scientific rationale for developing potent RBD-based vaccines targeting COVID-19." npj Vaccines **6**(1).

Uraki, R., et al. (2022). "Characterization of SARS-CoV-2 Omicron BA.4 and BA.5 isolates in rodents." Nature **612**(7940): 540-545.

Walsh, E. E., et al. (2020). "Safety and Immunogenicity of Two RNA-Based Covid-19 Vaccine Candidates." New England Journal of Medicine **383**(25): 2439-2450.

Reviewers' Comments:

Reviewer #2:

Remarks to the Author:

In fig 4. The authors present new ICS data to show antigen-specific T cells induced by gamma-adapted versus ancestral vaccine, which is appreciated. However, the representative staining (sFig4) are not convincing. The gates are partially capturing the cytokine-negative population, so it appears that the gates need to be moved. Was an FMO used to set the gates? If so, please include this in the representative plots.

The statement that gamma RBD elicited a mixed Th1/Th2 profile in contrast to ancestral RBD is still not well-supported by data. It is appreciated that the statistical comparisons are now (also) made between the gamma-adapted and ancestral vaccine. However, the authors found no statistically significant differences between the two. Also, if it was really the case that gamma RBD elicited a mixed Th1/Th2 profile in contrast to ancestral RBD, the authors should discuss underlying mechanisms for this. After all, the only difference between the two groups is in the antigen (RBD) and not in the adjuvant (alum) and since alum is known to elicit Th2 responses when combined with other protein subunit antigens and tested in mice, why would it be different with gamma-adapted RBD?

Point-by-point response to the reviewers' comments

Please find below a point-by-point response to the reviewers' comments. Reviewers' comments are written in black and responses in blue.

REVIEWER COMMENTS

Reviewer #2 (Remarks to the Author):

In fig 4. The authors present new ICS data to show antigen-specific T cells induced by gamma-adapted versus ancestral vaccine, which is appreciated. However, the representative staining (sFig4) are not convincing. The gates are partially capturing the cytokine-negative population, so it appears that the gates need to be moved. Was an FMO used to set the gates? If so, please include this in the representative plots.

Yes, we used an FMO control (cells labeled only with the viability dye and anti-CD4 and CD8 antibodies) to set the gates. As suggested by the reviewer and to further reduce the capture of the cytokine-negative population, we slightly moved the gates up to obtain < 0.08% of cytokine positive population in the FMO control. Representative dot plots of the FMO control are shown in Supplementary Figure 4 of the revised version of the manuscript. In addition, we annexed the dot plots from all samples at the end of this document. In the new revised version of the manuscript, new ICS data was replaced in the Figure 4a and new analysis of polyfunctional T cells is shown in Supplementary Figure 6 with similar results.

Results section was slightly modified according to this data:

Lines 206-212: *“Cellular immune responses were analyzed in splenocytes from immunized animals by intracellular flow cytometry. The Gamma RBD-based vaccine induced a significant increase in the frequencies of Gamma RBD-specific IFN- γ -, TNF- α - and IL-2-producing CD4⁺ T cells after restimulation with the Gamma antigen while the ancestral version of the vaccine failed to induce IFN- γ in CD4⁺ T cells but could induce TNF- α and IL-2 after restimulation (Fig. 4a, upper panel). Both vaccines had a similar performance at inducing antigen specific CD8⁺ T cells (Fig. 4a, lower panel).”*

Lines 217-220 (results section): *“In addition, analysis of polyfunctional T cells showed that responses elicited by the Gamma RBD based vaccine had a predominance of single positive TNF- α ⁺ cells and specific double-positive (IFN- γ ⁺ and TNF- α ⁺) cytokine producing CD4⁺ T cells (Supplementary Fig. 6).”*

The statement that gamma RBD elicited a mixed Th1/Th2 profile in contrast to ancestral RBD is still not well-supported by data. It is appreciated that the statistical comparisons are now (also) made between the gamma-adapted and ancestral vaccine. However, the authors found no statistically significant differences between the two. Also, if it was really the case that gamma RBD elicited a mixed Th1/Th2 profile in contrast to ancestral RBD, the authors should discuss underlying mechanisms for this. After all, the only difference between the two groups is in the antigen (RBD) and not in the adjuvant (alum) and since alum is known to elicit Th2 responses when combined with other protein subunit antigens and tested in mice, why would it be different with gamma-adapted RBD?

Regarding the results showing the induction of T helper cell immune responses by the two vaccine formulations in the Figure 4b. The Ag specific T helper immune response of each vaccine formulation was determined by evaluating the cytokines produced upon Ag stimulation, and this means the difference between cells stimulated with Ag or not. Since,

the gamma RBD vaccine formulation induces the production of significant amounts of IFN- γ and IL-4 upon Ag stimulation (using 2 different techniques/assays like ICS and multiplex) we called this response Th1-Th2. Of note, we confirmed in humans in a recently published phase 1 clinical trial that gamma RBD as booster increases the frequency of Ag specific IFN- γ and IL-4 producing T cells by ELISpot (Pasquevich, Coria et al. 2023).

In this work, Ag specific IFN- γ CD4⁺T cells were not significantly induced with the Ancestral vaccine (measured by ICS) neither significant IFN- γ levels were seen in the supernatants by multiplex assay. After immunization with the Ancestral RBD vaccine, predominantly Th2 related cytokines were significantly produced (IL-2, IL-6, IL-13, IL-4 and IL-5, Fig. 4b) after stimulation with the antigen in comparison to unstimulated cells, therefore we stated that the immune profile of the ancestral RBD based vaccine formulation was predominantly Th2.

As suggested by the reviewer and to further clarify this, we modified the results section better explaining which significant differences were found in comparison to unstimulated cells or between groups. We modified the statement about differences between vaccines. We also modified the related statements from abstract and discussion sections.

Lines 222-232 (results section): *“Evaluation of Th cytokine profile was performed by multiplex assay in supernatants from splenocytes of immunized mice stimulated with Ancestral or Gamma RBD antigen. Results revealed that the Gamma RBD vaccine induced cytokines associated with Ag-specific Th1 and Th2 responses with production of IFN- γ , TNF- α , IL-2, IL-6, IL-10, IL-4 and IL-5 in comparison to unstimulated cells (Fig. 4b). After immunization with the Ancestral RBD vaccine, predominantly Th2 related cytokines were significantly produced (IL-2, IL-6, IL-13, IL-4 and IL-5) upon Ag stimulation in comparison to unstimulated cells (Fig. 4b). Of note, significant differences between vaccine groups were only found in IL-5 and IL-13 production. Interestingly, Th17, Th9 and Th22 related cytokines (IL-17A/F, IL-9 and IL-22) were induced by the Gamma RBD + Alum formulation.*

These results demonstrated that both vaccines formulations elicited T cell immune responses with induction of Ag-specific cytokine producing CD8⁺ and CD4⁺ T cells.”

Lines 418-419 (discussion section): *“Both vaccine formulations elicited CD4⁺ and CD8⁺ T cell responses.”*

Regarding your last comment on the difference in the vaccine formulations, it is well known that the adjuvant is very important to drive the elicited T helper immune response but the Ag itself in the formulation also have a role in the final output of the immune response. It is very difficult to know exactly the outcome of the immune responses a priori and since yet there is a lot of empiricism in the final immune response elicited by a particular vaccine formulation, even using a very well-known adjuvant like Alum (Pulendran, S. Arunachalam et al. 2021). Besides, we have demonstrated in this work Gamma and ancestral RBD have different amino acid sequences that originate different T cell epitopes.

In addition to concerns of reviewer 2 on the ICS data, we note the difference in events captured between plots in Supplementary Figure 4, which should be clarified or ideally all plots should show the same number of events.

Regarding the editorial concern about different event numbers in the representative dot plots, these differences are because we had selected the more representative dot plots

of each condition and each cytokine for each plot, but they did not come from to the same sample (file ID numbers are different). In the revised Supplementary Figure 4 we used representative dot plots from the same sample (same condition and mouse), this way they show the same number of events.

Pasquevich, K. A., et al. (2023). "Safety and immunogenicity of a SARS-CoV-2 Gamma variant RBD-based protein adjuvanted vaccine used as booster in healthy adults." Nat Commun **14**(1): 4551.

Pulendran, B., et al. (2021). "Emerging concepts in the science of vaccine adjuvants." Nature Reviews Drug Discovery **20**(6): 454-475.

FMO control (only include Zombie Acqua dye and anti-CD4 and CD8 antibodies.)

FMO_G3_G03.fcs
CD4+
29557

FMO_G3_G03.fcs
CD4+
29557

FMO_G3_G03.fcs
CD4+
29557

FMO_G3_G03.fcs
CD8+
9408

FMO_G3_G03.fcs
CD8+
9408

FMO_G3_G03.fcs
CD8+
9408

RPMI 13-2_F2_F02.fcs
CD4+
48777

RPMI 13-2_F2_F02.fcs
CD4+
48777

RPMI 13-2_F2_F02.fcs
CD4+
48777

RPMI 13-2_F2_F02.fcs
CD8+
20863

RPMI 13-2_F2_F02.fcs
CD8+
20863

RPMI 13-2_F2_F02.fcs
CD8+
20863

RPMI 13-2_C2_C02.fcs
CD4+
49030

RPMI 13-2_C2_C02.fcs
CD4+
49030

RPMI 13-2_C2_C02.fcs
CD4+
49030

RPMI 13-2_C2_C02.fcs
CD8+
23197

RPMI 13-2_C2_C02.fcs
CD8+
23197

RPMI 13-2_C2_C02.fcs
CD8+
23197

RPMI 13-2_E2_E02.fcs
CD4+
34319

RPMI 13-2_E2_E02.fcs
CD4+
34319

RPMI 13-2_E2_E02.fcs
CD4+
34319

RPMI 13-2_E2_E02.fcs
CD8+
17675

RPMI 13-2_E2_E02.fcs
CD8+
17675

RPMI 13-2_E2_E02.fcs
CD8+
17675

RPMI 13-2_B2_B02.fcs
CD4+
43483

RPMI 13-2_B2_B02.fcs
CD4+
43483

RPMI 13-2_B2_B02.fcs
CD4+
43483

RPMI 13-2_B2_B02.fcs
CD8+
22214

RPMI 13-2_B2_B02.fcs
CD8+
22214

RPMI 13-2_B2_B02.fcs
CD8+
22214

RPMI 13-2_D2_D02.fcs
CD4+
39649

RPMI 13-2_D2_D02.fcs
CD4+
39649

RPMI 13-2_D2_D02.fcs
CD4+
39649

RPMI 13-2_D2_D02.fcs
CD8+
20258

RPMI 13-2_D2_D02.fcs
CD8+
20258

RPMI 13-2_D2_D02.fcs
CD8+
20258

RPMI 13-2_A2_A02.fcs
CD4+
42896

RPMI 13-2_A2_A02.fcs
CD4+
42896

RPMI 13-2_A2_A02.fcs
CD4+
42896

RPMI 13-2_A2_A02.fcs
CD8+
22345

RPMI 13-2_A2_A02.fcs
CD8+
22345

RPMI 13-2_A2_A02.fcs
CD8+
22345

RPMI 13-1_F1_F01.fcs
CD4+
68996

RPMI 13-1_F1_F01.fcs
CD4+
68996

RPMI 13-1_F1_F01.fcs
CD4+
68996

RPMI 13-1_F1_F01.fcs
CD8+
32466

RPMI 13-1_F1_F01.fcs
CD8+
32466

RPMI 13-1_F1_F01.fcs
CD8+
32466

RPMI 13-1_C1_C01.fcs
CD4+
48353

RPMI 13-1_C1_C01.fcs
CD4+
48353

RPMI 13-1_C1_C01.fcs
CD4+
48353

RPMI 13-1_C1_C01.fcs
CD8+
25775

RPMI 13-1_C1_C01.fcs
CD8+
25775

RPMI 13-1_C1_C01.fcs
CD8+
25775

RPMI 13-1_F1_F01.fcs
CD4+
68996

RPMI 13-1_F1_F01.fcs
CD4+
68996

RPMI 13-1_F1_F01.fcs
CD4+
68996

RPMI 13-1_F1_F01.fcs
CD8+
32466

RPMI 13-1_F1_F01.fcs
CD8+
32466

RPMI 13-1_F1_F01.fcs
CD8+
32466

RPMI 13-1_C1_C01.fcs
CD4+
48353

RPMI 13-1_C1_C01.fcs
CD4+
48353

RPMI 13-1_C1_C01.fcs
CD4+
48353

RPMI 13-1_C1_C01.fcs
CD8+
25775

RPMI 13-1_C1_C01.fcs
CD8+
25775

RPMI 13-1_C1_C01.fcs
CD8+
25775

RPMI 13-1_D1_D01.fcs
CD4+
60560

RPMI 13-1_D1_D01.fcs
CD4+
60560

RPMI 13-1_D1_D01.fcs
CD4+
60560

RPMI 13-1_D1_D01.fcs
CD8+
33739

RPMI 13-1_D1_D01.fcs
CD8+
33739

RPMI 13-1_D1_D01.fcs
CD8+
33739

RPMI 13-1_A1_A01.fcs
CD4+
42733

RPMI 13-1_A1_A01.fcs
CD4+
42733

RPMI 13-1_A1_A01.fcs
CD4+
42733

RPMI 13-1_A1_A01.fcs
CD8+
24227

RPMI 13-1_A1_A01.fcs
CD8+
24227

RPMI 13-1_A1_A01.fcs
CD8+
24227

Excluded data- bad quality of sample

Wuhan 13-2_F5_F05.fcs
CD4+
23324

Wuhan 13-2_F5_F05.fcs
CD4+
23324

Wuhan 13-2_F5_F05.fcs
CD4+
23324

Wuhan 13-2_F5_F05.fcs
CD8+
14764

Wuhan 13-2_F5_F05.fcs
CD8+
14764

Wuhan 13-2_F5_F05.fcs
CD8+
14764

Wuhan 13-2_C5_C05.fcs
CD4+
39163

Wuhan 13-2_C5_C05.fcs
CD4+
39163

Wuhan 13-2_C5_C05.fcs
CD4+
39163

Wuhan 13-2_C5_C05.fcs
CD8+
19155

Wuhan 13-2_C5_C05.fcs
CD8+
19155

Wuhan 13-2_C5_C05.fcs
CD8+
19155

Wuhan 13-2_E5_E05.fcs
CD4+
52606

Wuhan 13-2_E5_E05.fcs
CD4+
52606

Wuhan 13-2_E5_E05.fcs
CD4+
52606

Wuhan 13-2_E5_E05.fcs
CD8+
25887

Wuhan 13-2_E5_E05.fcs
CD8+
25887

Wuhan 13-2_E5_E05.fcs
CD8+
25887

Wuhan 13-2_B5_B05.fcs
CD4+
27071

Wuhan 13-2_B5_B05.fcs
CD4+
27071

Wuhan 13-2_B5_B05.fcs
CD4+
27071

Wuhan 13-2_B5_B05.fcs
CD8+
15894

Wuhan 13-2_B5_B05.fcs
CD8+
15894

Wuhan 13-2_B5_B05.fcs
CD8+
15894

Wuhan 13-2_D5_D05.fcs
CD4+
27064

Wuhan 13-2_D5_D05.fcs
CD4+
27064

Wuhan 13-2_D5_D05.fcs
CD4+
27064

Wuhan 13-2_D5_D05.fcs
CD8+
16457

Wuhan 13-2_D5_D05.fcs
CD8+
16457

Wuhan 13-2_D5_D05.fcs
CD8+
16457

Wuhan 13-2_A5_A05.fcs
CD4+
61531

Wuhan 13-2_A5_A05.fcs
CD4+
61531

Wuhan 13-2_A5_A05.fcs
CD4+
61531

Wuhan 13-2_A5_A05.fcs
CD8+
31959

Wuhan 13-2_A5_A05.fcs
CD8+
31959

Wuhan 13-2_A5_A05.fcs
CD8+
31959

Wuhan 13-1_F4_F04.fcs
CD4+
64755

Wuhan 13-1_F4_F04.fcs
CD4+
64755

Wuhan 13-1_F4_F04.fcs
CD4+
64755

Wuhan 13-1_F4_F04.fcs
CD8+
32684

Wuhan 13-1_F4_F04.fcs
CD8+
32684

Wuhan 13-1_F4_F04.fcs
CD8+
32684

Wuhan 13-1_C4_C04.fcs
CD4+
35018

Wuhan 13-1_C4_C04.fcs
CD4+
35018

Wuhan 13-1_C4_C04.fcs
CD4+
35018

Wuhan 13-1_C4_C04.fcs
CD8+
19534

Wuhan 13-1_C4_C04.fcs
CD8+
19534

Wuhan 13-1_C4_C04.fcs
CD8+
19534

Wuhan 13-1_E4_E04.fcs
CD4+
32968

Wuhan 13-1_E4_E04.fcs
CD4+
32968

Wuhan 13-1_E4_E04.fcs
CD4+
32968

Wuhan 13-1_E4_E04.fcs
CD8+
18600

Wuhan 13-1_E4_E04.fcs
CD8+
18600

Wuhan 13-1_E4_E04.fcs
CD8+
18600

Wuhan 13-1_B4_B04.fcs
CD4+
42414

Wuhan 13-1_B4_B04.fcs
CD4+
42414

Wuhan 13-1_B4_B04.fcs
CD4+
42414

Wuhan 13-1_B4_B04.fcs
CD8+
22031

Wuhan 13-1_B4_B04.fcs
CD8+
22031

Wuhan 13-1_B4_B04.fcs
CD8+
22031

Wuhan 13-1_D4_D04.fcs
CD4+
45256

Wuhan 13-1_D4_D04.fcs
CD4+
45256

Wuhan 13-1_D4_D04.fcs
CD4+
45256

Wuhan 13-1_D4_D04.fcs
CD8+
26074

Wuhan 13-1_D4_D04.fcs
CD8+
26074

Wuhan 13-1_D4_D04.fcs
CD8+
26074

Wuhan 13-1_A4_A04.fcs
CD4+
39046

Wuhan 13-1_A4_A04.fcs
CD4+
39046

Wuhan 13-1_A4_A04.fcs
CD4+
39046

Wuhan 13-1_A4_A04.fcs
CD8+
24192

Wuhan 13-1_A4_A04.fcs
CD8+
24192

Wuhan 13-1_A4_A04.fcs
CD8+
24192

Gamma 13-2_F8_F08.fcs
CD4+
74968

Gamma 13-2_F8_F08.fcs
CD4+
74968

Gamma 13-2_F8_F08.fcs
CD4+
74968

Gamma 13-2_F8_F08.fcs
CD8+
27367

Gamma 13-2_F8_F08.fcs
CD8+
27367

Gamma 13-2_F8_F08.fcs
CD8+
27367

Gamma 13-2_C8_C08.fcs
CD4+
58135

Gamma 13-2_C8_C08.fcs
CD4+
58135

Gamma 13-2_C8_C08.fcs
CD4+
58135

Gamma 13-2_C8_C08.fcs
CD8+
23827

Gamma 13-2_C8_C08.fcs
CD8+
23827

Gamma 13-2_C8_C08.fcs
CD8+
23827

Gamma 13-2_E8_E08.fcs
CD4+
50827

Gamma 13-2_E8_E08.fcs
CD4+
50827

Gamma 13-2_E8_E08.fcs
CD4+
50827

Gamma 13-2_E8_E08.fcs
CD8+
24158

Gamma 13-2_E8_E08.fcs
CD8+
24158

Gamma 13-2_E8_E08.fcs
CD8+
24158

Gamma 13-2_B8_B08.fcs
CD4+
26542

Gamma 13-2_B8_B08.fcs
CD4+
26542

Gamma 13-2_B8_B08.fcs
CD4+
26542

Gamma 13-2_B8_B08.fcs
CD8+
13783

Gamma 13-2_B8_B08.fcs
CD8+
13783

Gamma 13-2_B8_B08.fcs
CD8+
13783

Gamma 13-2_D8_D08.fcs
CD4+
23589

Gamma 13-2_D8_D08.fcs
CD4+
23589

Gamma 13-2_D8_D08.fcs
CD4+
23589

Gamma 13-2_D8_D08.fcs
CD8+
12823

Gamma 13-2_D8_D08.fcs
CD8+
12823

Gamma 13-2_D8_D08.fcs
CD8+
12823

Gamma 13-2_A8_A08.fcs
CD4+
50561

Gamma 13-2_A8_A08.fcs
CD4+
50561

Gamma 13-2_A8_A08.fcs
CD4+
50561

Gamma 13-2_A8_A08.fcs
CD8+
21541

Gamma 13-2_A8_A08.fcs
CD8+
21541

Gamma 13-2_A8_A08.fcs
CD8+
21541

Gamma 13-1_F7_F07.fcs
CD4+
46856

Gamma 13-1_F7_F07.fcs
CD4+
46856

Gamma 13-1_F7_F07.fcs
CD4+
46856

Gamma 13-1_F7_F07.fcs
CD8+
24281

Gamma 13-1_F7_F07.fcs
CD8+
24281

Gamma 13-1_F7_F07.fcs
CD8+
24281

Gamma 13-1_C7_C07.fcs
CD4+
43356

Gamma 13-1_C7_C07.fcs
CD4+
43356

Gamma 13-1_C7_C07.fcs
CD4+
43356

Gamma 13-1_C7_C07.fcs
CD8+
23172

Gamma 13-1_C7_C07.fcs
CD8+
23172

Gamma 13-1_C7_C07.fcs
CD8+
23172

Gamma 13-1_E7_E07.fcs
CD4+
29759

Gamma 13-1_E7_E07.fcs
CD4+
29759

Gamma 13-1_E7_E07.fcs
CD4+
29759

Gamma 13-1_E7_E07.fcs
CD8+
17121

Gamma 13-1_E7_E07.fcs
CD8+
17121

Gamma 13-1_E7_E07.fcs
CD8+
17121

Gamma 13-1_B7_B07.fcs
CD4+
33076

Gamma 13-1_B7_B07.fcs
CD4+
33076

Gamma 13-1_B7_B07.fcs
CD4+
33076

Gamma 13-1_B7_B07.fcs
CD8+
17690

Gamma 13-1_B7_B07.fcs
CD8+
17690

Gamma 13-1_B7_B07.fcs
CD8+
17690

Gamma 13-1_D7_D07.fcs
CD4+
41731

Gamma 13-1_D7_D07.fcs
CD4+
41731

Gamma 13-1_D7_D07.fcs
CD4+
41731

Gamma 13-1_D7_D07.fcs
CD8+
23241

Gamma 13-1_D7_D07.fcs
CD8+
23241

Gamma 13-1_D7_D07.fcs
CD8+
23241

Gamma 13-1_A7_A07.fcs
CD4+
26456

Gamma 13-1_A7_A07.fcs
CD4+
26456

Gamma 13-1_A7_A07.fcs
CD4+
26456

Gamma 13-1_A7_A07.fcs
CD8+
14335

Gamma 13-1_A7_A07.fcs
CD8+
14335

Gamma 13-1_A7_A07.fcs
CD8+
14335